# A POLD3/BLM dependent pathway handles DSBs in transcribed chromatin upon excessive RNA:DNA hybrid accumulation

S. Cohen [1,4], A. Guenolé[1,4], I. Lazar [1,4], A. Marnef [1], T. Clouaire [1], D. V. Vernekar[2], N. Puget[1], V. Rocher[1], C. Arnould[1], M. Aguirrebengoa[1], M. Genais[1], N. Firmin[1], R. A. Shamanna[3], R. Mourad[1], V. A. Bohr[3], V. Borde [2] & G. Legube [1✉]

Transcriptionally active loci are particularly prone to breakage and mounting evidence suggests that DNA Double-Strand Breaks arising in active genes are handled by a dedicated repair pathway, Transcription-Coupled DSB Repair (TC-DSBR), that entails R-loop accumulation and dissolution. Here, we uncover a function for the Bloom RecQ DNA helicase (BLM) in TC-DSBR in human cells. BLM is recruited in a transcription dependent-manner at DSBs where it fosters resection, RAD51 binding and accurate Homologous Recombination repair. However, in an R-loop dissolution-deficient background, we find that BLM promotes cell death. We report that upon excessive RNA:DNA hybrid accumulation, DNA synthesis is enhanced at DSBs, in a manner that depends on BLM and POLD3. Altogether our work unveils a role for BLM at DSBs in active chromatin, and highlights the toxic potential of RNA:DNA hybrids that accumulate at transcription-associated DSBs.

---

[1] MCD, Centre de Biologie Intégrative (CBI), CNRS, Université de Toulouse, UT3, Toulouse, France. [2] Institut Curie, Université PSL, Sorbonne Université, CNRS UMR3244, Dynamics of Genetic Information, Paris, France. [3] Section on DNA Repair, National Institute on Aging, National Institutes of Health, Baltimore, MD, USA. [4] These authors contributed equally: S. Cohen, A. Guenolé, I. Lazar. ✉email: gaelle.legube@univ-tlse3.fr

DNA double-strand breaks (DSBs) are harmful lesions that occur in the genome following exposure to various environmental sources such as radiation or chemotherapy, and that also arise on a regular basis due to cell metabolic activity including during replication or the release of topological stress. Genome-wide sequencing-based analyses unveiled that endogenous DSBs primarily occur in genomic loci prone to form secondary structures, such as G-quadruplexes (G4), transcribed regions, or CTCF-bound loci (for instance[1–6] reviewed in ref. [7]).

While DSB repair pathways, including Non-Homologous End Joining (NHEJ) and Homologous Recombination (HR), have been well characterized (reviewed in ref. [8]), recent evidence suggests that repairing DSBs in RNA Polymerase II (RNAPII)-transcribed loci requires additional mechanisms, collectively referred to as Transcription-coupled DSB repair (TC-DSBR)[9–11]. In post-replicative cells, DSBs located in transcribed loci (TC-DSBs) are channeled to a specific homologous recombination repair pathway while in G1 these damages tend to persist and cluster[12,13]. The G2 arm of TC-DSBR, also known as TAHRR (for transcription-associated homologous recombination repair)[14], involves RNA:DNA hybrids accumulation and resolution in a Senataxin (SETX), XPG/ERCC5, DDX1, and EXOSC10-dependent manner[14–18]. The mechanisms that account for RNA:DNA hybrids accumulation and their exact nature (either R-loops composed of RNA:DNA hybrids and a displaced ssDNA, or RNA:DNA hybrids formed through hybridization of an RNA to the resected DNA strand) are still under investigation. They have been proposed to arise either due to transcriptional arrest induced at damaged active genes[19–21] (reviewed in refs. [7,22]) or DNA end-mediated de novo transcription by RNAPII or RNAPIII[23,24]. Moreover, RNA:DNA hybrid accumulation requires the miRNA processing enzyme DROSHA[25] as well as resection[14,26] (for review, see ref. [22]). RNA:DNA hybrids further contribute to recruit RAD52[14] and BRCA2[26], and to promote HR by forming DR-loops on donor DNA[27]. Their resolution is mandatory for completion of homologous recombination[14,16,18]. This is highlighted by the strong survival defect upon depletion of the RNA:DNA hybrid helicase SETX, which is only observed when DSBs are induced in active loci, and not randomly distributed across the genome[18].

The Bloom Syndrome helicase (BLM) is a 3' to 5' DNA helicase mutated in Bloom Syndrome which is a genetic disorder associated with an increased risk of cancer, sun-induced chronic erythema, impaired fertility, and immune deficiency. BLM displays pleiotropic functions in response to DSBs. First, it contributes to HR via the dissolution of double Holliday junctions[28–30], heteroduplex rejection[31] and by promoting long-range resection[28,31–37] (for review, see refs. [38,39]). Second, BLM also displays anti-recombinogenic properties, since (i) it promotes 53BP1 (an anti-resection factor) foci assembly and acts together with 53BP1 and RIF1 to protect against RBBP8/CTIP-dependent long-range deletions[40–42], (ii) it disrupts RAD51 filament assembled on ssDNA and inhibits D-loop formation[43] and (iii) its depletion rescues RAD51 loading in an HR-impaired, BRCA1Δ11 mutant, background[44]. Finally, BLM (or its yeast counterpart Sgs1) was also found to contribute to Break-Induced Replication (BIR)[45] and BIR-like pathways such as during Alternative Lengthening of Telomeres (ALT), by promoting ALT-associated PML body (APB) formation and BIR-intermediates resolution[46–52]. Of interest, previous work revealed that sister chromatid exchange (SCEs) breakpoints observed in BLM-deficient cells are biased toward transcribed genes[53], raising the interesting possibility that BLM may be a bona fide component of TC-DSBR.

Here, in order to investigate the function of BLM during TC-DSBR we describe the analysis of genome-wide distribution of BLM, at a high resolution and at multiple DSBs induced simultaneously in transcribed and un-transcribed loci in human cells. We find that the recruitment of BLM is biased towards DSBs in RNAPII-bound loci and G4-prone loci and depends on transcriptional activity. At these DSBs, BLM contributes to resection and RAD51 recruitment, ensuring faithful repair. However, we find that in a context of excessive RNA:DNA hybrid accumulation that impairs HR[18], BLM promotes cell death. POLD3 and BLM-dependent DNA synthesis takes place at DSBs upon impaired RNA:DNA hybrid dissolution, increasing translocation frequency and cell death. Genomic analyses of pancreatic cancer patient samples reveal that expression levels of SETX and BLM correlate with the occurrence of mutational signatures previously associated to BIR. Altogether our data suggest that BLM plays a pleiotropic role at DSBs in transcriptionally active loci, by promoting HR under normal circumstances, and by fostering an alternative, POLD3-dependent, cytotoxic repair pathway, in cells deficient for RNA:DNA hybrid dissolution.

## Results

**BLM is recruited to transcription-coupled DSBs.** In order to get insights into the function of BLM during DSB repair, we analyzed its genome-wide distribution by ChIP-seq using the DIvA cell system (DSB Inducible by AsiSI)[54]. This cell line expresses a restriction enzyme (AsiSI) fused to the ligand-binding domain of the estrogen receptor. Following treatment with 4-hydroxytamoxifen (4OHT), the fusion enzyme relocates to the nucleus, where it homogeneously induces, multiple DSBs at annotated positions located both in RNAPII-bound (promoters, genes bodies) and unbound (silent genes, intergenic) loci[12,55]. ChIP-seq analysis post damage indicated that BLM is recruited at the vicinity of DSBs (Fig. 1a, two well-induced DSBs in transcribed loci). Average BLM profile and heatmaps at all DSBs showed that BLM spreads on roughly 5–10 kb around DSBs and peaks at a position where γH2AX is depleted (Supplementary Fig. 1a, Fig. 1b). These data therefore indicate that the BLM helicase is distributed on restricted domains around DSBs unlike the megabase-wide spreading observed for γH2AX.

Individual inspection of DSBs revealed that for equivalent cleavage (determined by BLESS[55]), some DSBs displayed increased BLM binding compared to others (example Supplementary Fig. 1b). Given the ability of BLM to bind G4, we further compared BLM binding with the available G4 mapping generated by BG4 immunoprecipitation in HaCaT cells[56]. Interestingly following damage induction, BLM tended to be recruited at loci enriched in G4 (Supplementary Fig. 1b). In order to further quantify this observation, and determine the genomic/epigenomic determinants that foster BLM recruitment at some DSBs compared to others we identified subsets of DSBs differentially enriched in BLM, as previously described[12,55]. To this aim, we used our previously published BLESS dataset[55] to account for potential unequal activity of AsiSI throughout the genome and computed the BLM/BLESS enrichment ratio for each of the 80 most cleaved DSBs. This approach allowed us to identify a subset of DSBs enriched in BLM (BLM-high) while some others are only poorly able to recruit BLM (BLM-low) (Supplementary Fig. 1c, d). On average, BLM-high DSBs were enriched in G4 compared to the BLM-low subset of DSBs (Supplementary Fig. 1e).

We then analyzed BLM binding with respect to transcriptional activity using ChIP-Seq datasets generated before break induction in the DIvA model against total RNA-Polymerase II (RNAPII) and phosphorylated RNAPII on serine 2 (RNAPII-S2P, associated with transcriptional elongation)[12,18]. The genes located at the immediate vicinity of BLM-high DSBs exhibited the typical pattern of actively transcribed genes, with an accumulation of

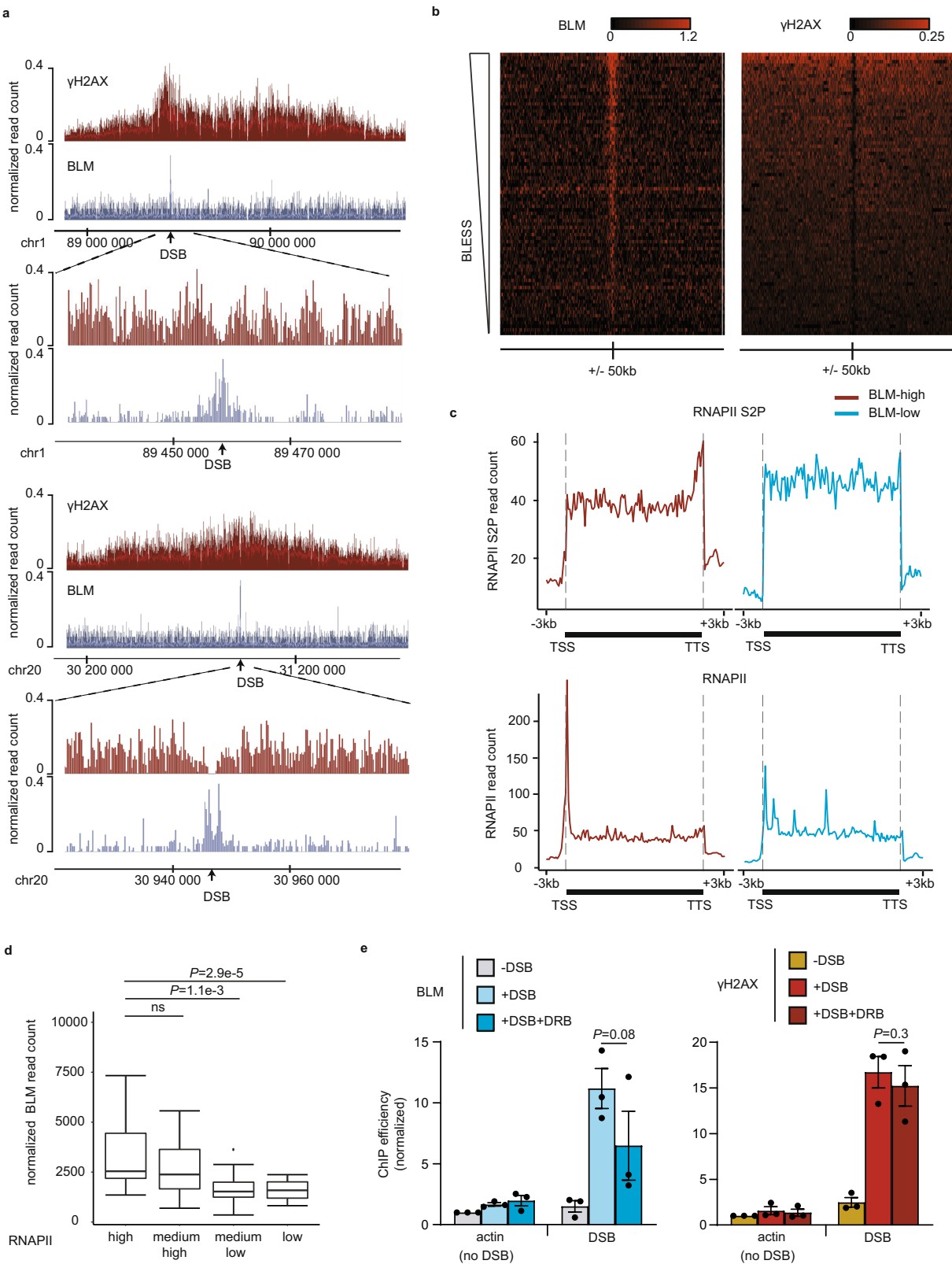

RNAPII at promoters and RNAPII-S2P at transcription termination site (TTS) in contrast to the genes lying near BLM-low DSBs (Fig. 1c). In agreement, loci showing a strong RNAPII and RNAPII-S2P signal before DSB induction, displayed higher BLM recruitment following DSB induction (Fig. 1d, Supplementary Fig. 1f, g). The trend for DSBs located in transcriptionally active

loci to recruit BLM was further confirmed by visual inspection of the profiles of BLM and RNAPII ChIP-seq as well as examining RNA-seq data at individual DSBs (Supplementary Fig. 1h). To determine the influence of prior transcriptional activity on BLM recruitment, we next performed BLM ChIP in the presence of 5,6-Dichlorobenzimidazole 1-β-D-ribofuranoside (DRB), a selective

**Fig. 1 Transcription-dependent BLM recruitment at DSBs. a** Genome browser screenshots representing γH2AX and BLM ChIP-seq signal after DSB induction (4 h) at two well-induced DSBs in the DIvA system (chr1:89458586 (upper panel) and chr20:30946312 (lower panel)). Magnifications are also showed. DSBs are indicated by arrows. **b** Heatmaps showing the γH2AX (right panel) and BLM (left panel) ChIP-seq signal on a 100 kb window centered around 80 DSB sites. **c** Average ChIP-seq profiles of RNAPII S2P (top panel) and Total RNAPII (bottom panel) across the closest genes lying near BLM-high (red) or BLM-low DSBs (blue). **d** Boxplot representing normalized BLM ChIP-seq read count on 5 kb around DSB after DSB induction on loci with high ($n = 20$), medium high ($n = 20$), medium low ($n = 20$), or low ($n = 20$) RNAPII occupancy before DSB induction (determined by ChIP-seq before DSB induction). Center line: median; Box limits: 1st and 3rd quartile; Whiskers: Maximum and minimum values; Points: outliers. P values, non-parametric two-sided Wilcoxon test. **e** BLM (blue) and γH2AX (red) ChIP efficiency (expressed as % of input immunoprecipitated) before (−DSB), after (+DSB for 4 h) DSB induction, and after DSB induction with prior inhibition (1 h pre-DSB induction) of transcription elongation (+DSB + DRB) at 80 bp from a DSB enriched in BLM (DSB1: chr22:38864101) and on a control site (actin; no-DSB). Mean and SEM are shown for $n = 3$ biologically independent experiments. P, paired t-test (two-sided).

inhibitor of RNAPII transcriptional elongation. BLM recruitment post damage was reduced in DRB pre-treated cells compared to untreated cells, while the γH2AX signal, and hence DSB induction, was unaffected (Fig. 1e), indicating that transcription activity fosters BLM recruitment upon DSB. Altogether these data indicate that DSBs occurring in RNAPII-transcribed loci (TC-DSBs) display an increased potential to recruit BLM.

**BLM functions in transcription-coupled DSB repair.** Given that BLM has been extensively involved in HR, we first investigated the consequence of cell cycle stage on BLM recruitment at DSBs. Kinetics using GFP-tagged BLM showed a recruitment of BLM at micro irradiation sites throughout the cell cycle, although enhanced in G2, cyclinA-positive cells, compared to G1 cells (Supplementary Fig. 2a). ChIP experiments in synchronized DIvA cells further confirmed enhanced binding at TC-DSB in G2 cells, although it was still detectable in G1 (Supplementary Fig. 2b). We next compared the binding of BLM with that of RAD51, previously reported by ChIP-seq in DIvA cells[12]. Notably, BLM strongly paralleled RAD51 distribution (Fig. 2a and aggregate profiles at BLM-high and BLM-low DSBs subsets Fig. 2b). Moreover, BLM-low sites displayed a low ability to recruit RAD51 (Fig. 2a, b), despite equivalent cleavage (BLESS signal, Fig. 2a). Of interest, mapping of RAD51 and BLM after longer exposure to AsiSI activity (24 h) showed considerably extended spreading of RAD51 and BLM (Fig. 2c), with RAD51 distribution again highly correlating with BLM distribution (Fig. 2d). Altogether these data suggest that BLM spreads on surrounding chromatin as resection progresses and as the RAD51 filament assembles.

In order to determine the function of BLM at TC-DSBs, we performed siRNA depletion (Supplementary Fig. 3a left panel and middle panel) and confirmed a clear drop in BLM recruitment on damaged chromatin (Supplementary Fig. 3a right panel) without disturbing the cell cycle (Supplementary Fig. 3b). Using a previously described assay that relies on the inability of restriction enzymes to cleave single-strand DNA[57], we found that BLM depletion triggers a reduction in DSB resection as expected (Fig. 2e). In agreement, BLM depletion also triggered a reduction in RAD51 foci formation (Fig. 2f) and in RAD51 binding at a TC-DSB (Fig. 2g). We next assessed the kinetics of TC-DSB repair upon BLM depletion. For this, we used the AID-DIvA cells, which enable rapid degradation of AsiSI upon auxin addition, and an assay allowing capture and quantification of unrepaired DSBs[12]. BLM knock down mildly delayed repair, a phenotype previously observed when depleting RAD51[12] (Supplementary Fig. 3c). We further assessed the fidelity of TC-DSBR in AID-DIvA cells taking advantage of the fact that upon accurate repair, the AsiSI restriction site is reconstituted post-auxin addition, and thus becomes available for a new round of cleavage by 4OHT treatment[58] (DSB + IAA + DSB). We observed decreased repair fidelity in BLM-depleted cells, compared to control cells

(Supplementary Fig. 3d). Of interest, ChIP against PARP1 revealed increased PARP1 recruitment at DSB upon BLM depletion (Supplementary Fig. 3e), suggesting enhanced Alt-NHEJ usage.

Altogether these data indicate that BLM is recruited at TC-DSBs in a manner that coincides with the progression of resection. At TC-DSBs, BLM fosters resection, RAD51 binding and accurate repair. Thus, BLM plays a key role in TC-DSBR to ensure faithful Transcription-Associated Homologous Recombination Repair (TAHRR).

**BLM promotes cell death in SETX-deficient cells following DSBs in transcribed loci.** We previously identified the SETX RNA:DNA helicase as another critical component involved in TC-DSBR, due to its ability to unwind RNA:DNA hybrids[18]. Like BLM, SETX is specifically recruited to DSBs induced in transcribed loci and not at DSBs induced in intergenic and silent genes/promoters[18]. Notably, SETX and BLM spanned similar regions surrounding DSBs (Fig. 3a) and the recruitment levels of both proteins significantly correlated at all DSBs (Fig. 3b). Given our above findings we hence further investigated the interplay between SETX and BLM in TC-DSBR. As previously shown[18] SETX depletion impaired cell survival following DSBs. Unexpectedly, BLM depletion partially rescued the cell death observed in SETX-deficient cells (Fig. 3c, Supplementary Fig. 4a).

Increased lethality upon DSB-induction in SETX-depleted cells was attributed to an impaired removal of DSB-induced RNA:DNA hybrids, precluding RAD51 foci assembly and the execution of HR[18]. Previous studies reported that resection at sites of damage is a prerequisite for RNA:DNA hybrid accumulation[26]. Given that BLM promotes long-range resection[28,31–37], BLM deficiency may rescue DSB-induced cell death in SETX-deficient cells by decreasing the ability of RNA:DNA hybrids to form *in cis* to DSBs, due to impaired resection. Indeed, BLM depletion triggered a reduction in resection in both SETX-proficient and deficient cells (Supplementary Fig. 4b) which could therefore impair RNA:DNA hybrid formation. Hence, we further investigated by DRIP-seq the consequence of BLM depletion on RNA:DNA hybrid accumulation in a SETX-deficient background. As expected, SETX depletion triggered an increase of RNA:DNA hybrids at DSBs (Fig. 3d, Supplementary Fig. 4c, d). Interestingly, however, BLM depletion did not reduce RNA:DNA hybrid formation at DSBs in SETX-depleted cells (Fig. 3d, Supplementary Fig. 4d). Therefore, despite decreased DSB resection upon BLM depletion (Supplementary Fig. 4b), RNA:DNA hybrids are still able to accumulate *in cis* to DSB. Upon prolonged DSB induction (24 h), ChIP-seq showed a clear spreading of RAD51, indicative of extended resection and nucleofilament assembly under these conditions (Figs. 2c and 3e, compare 4 h to 24 h). However, DRIP-seq at 4 h and 24 h after DSB induction showed little effect on RNA:DNA hybrid distribution *in cis* to DSBs (Fig. 3e). Altogether, these data

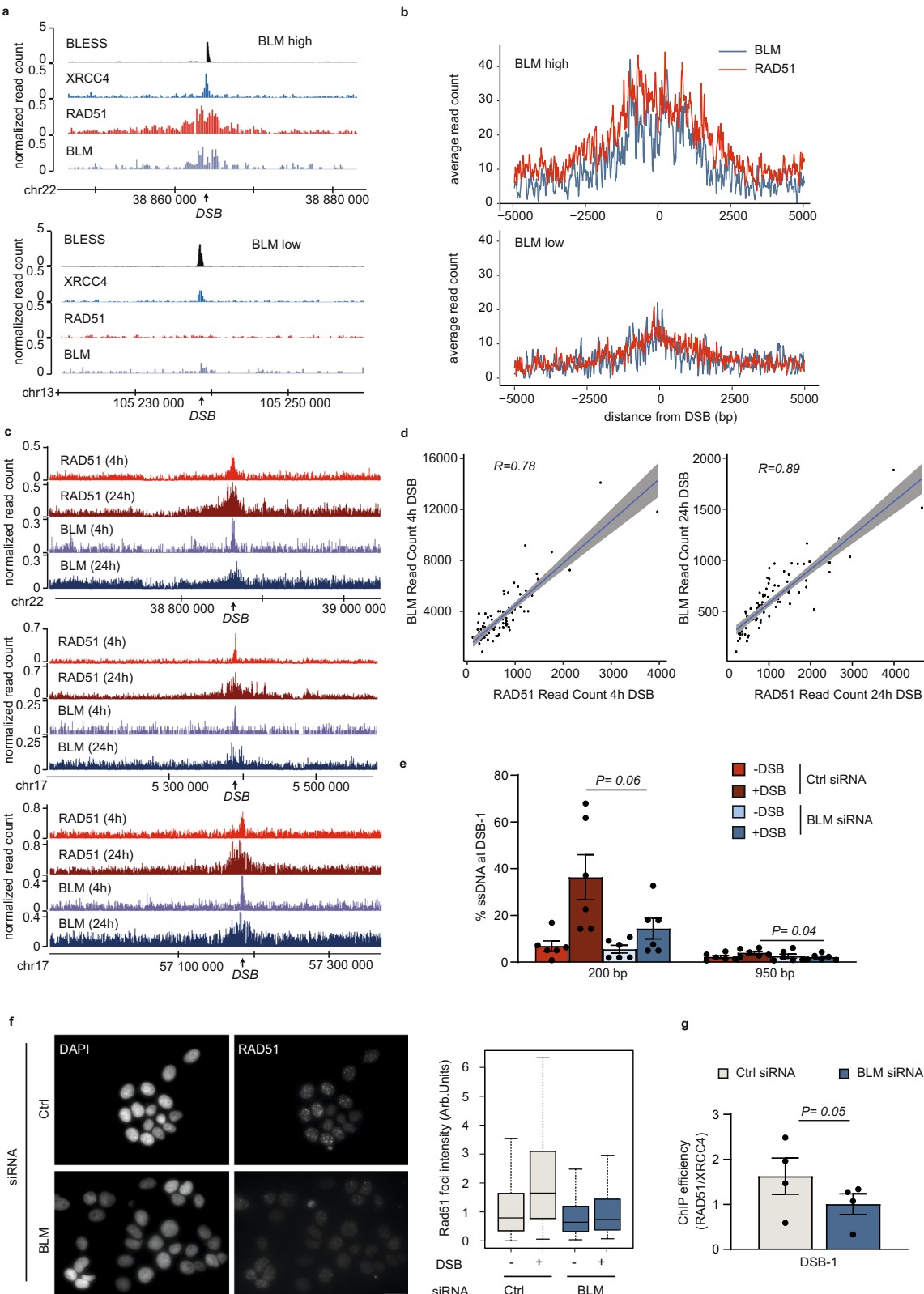

suggest that RNA:DNA hybrid formation *in cis* to DSBs neither follows the resection pattern, nor is affected by a decreased availability of resected DNA ends.

In order to further investigate whether decreased resection by itself could rescue the DSB-induced lethality observed in SETX-deficient cells, we depleted CTIP by siRNA, which, as previously reported, led to decreased ssDNA formation (Supplementary

Fig. 4e). CTIP depletion, unlike BLM depletion, did not rescue the DSB-induced lethality observed in SETX-deficient cells (Supplementary Fig. 4f), which further supports a role of BLM in promoting cell death upon excessive RNA:DNA hybrid accumulation independently from its role in promoting resection.

We previously reported that defective R-loop removal in SETX-depleted cells impaired RAD51 foci formation. We therefore tested

**Fig. 2 BLM recruitment at TC-DSBs fosters resection, RAD51 loading, and repair fidelity. a** Genome browser screenshots representing BLESS, XRCC4, RAD51, and BLM ChIP-seq signal at a BLM-high DSB (upper panel; chr22:38864101) and a BLM-low DSB (lower panel; chr13:105238551) (4 h of DSB induction). DSBs are indicated by arrows. **b** Average profiles of BLM (blue) and RAD51 (red) on BLM-high and BLM-low DSB ($n = 20$) sites after DSB induction (10 kb window). **c** Genome browser screenshot representing RAD51 and BLM ChIP-seq signal at 4 h and 24 h after DSB induction at three DSBs (arrows) (located on chr22:38864101, chr17:5390220, and chr17:57184296). **d** Scatter plots representing BLM and RAD51 ChIP-seq read counts at the best eighty DSBs on a 10 kb window at 4 h (left panel) and on a 40 kb at 24 h (right panel) after DSB induction. R (Pearson) is indicated. Error in gray represents the regression confidence as obtained by the geom_smooth function of ggplot2 (linear model). **e** Resection assay showing the percentage of single-strand DNA (% ssDNA) in control (red) and BLM (blue) siRNA-depleted DIvA cells at two distances from the DSB-1 (chr22:38864101, bound by BLM) as indicated. Mean and SEM are shown for $n = 6$ biologically independent experiments. P, paired t-test (two-sided). **f** (Left) RAD51 immunostaining performed in control (Ctrl) and BLM siRNA depleted DIvA cells after DSB induction for 4 h. Scale bar 20 μM. (Right) Quantification of RAD51 foci intensity before (−) and after (+) DSB induction in control (Ctrl) and BLM siRNA depleted DIvA. Ctrl-DSB, $n = 248$ cells; Ctrl + DSB, $n = 243$ cells; BLM-DSB, $n = 301$ cells; BLM + DSB, $n = 142$ cells. Center line: median; Box limits: 1st and 3rd quartile; Whiskers: Maximum and minimum values without outliers. A representative experiment is shown. **g** ChIP efficiency (RAD51/XRCC4) at a TC-DSB (DSB-1 chr22:38864101) in control (gray) and BLM (blue) siRNA depleted DIvA cells after DSB induction for 4 h. Mean and SEM are shown for $n = 4$ biologically independent experiments. P, paired t-test (two-sided).

the impact of BLM depletion on RAD51 nucleofilament assembly. We found that BLM loss did not rescue RAD51 foci formation in SETX-depleted cells (Fig. 3f), in agreement with the sustained RNA:DNA hybrid levels in the double BLM/SETX-deficient background (Fig. 3d).

Altogether these data indicate that (i) RNA:DNA hybrid formation *in cis* to DSBs induced in active loci is not a direct consequence of ssDNA generation, since their distribution do not follow resection and since they are not reduced upon impaired resection and that (ii) BLM promotes cell toxicity upon impaired RNA:DNA hybrid removal, downstream of their accumulation and in a manner that is independent from its role in long-range resection.

**BLM promotes DNA synthesis in a SETX-deficient background.** In order to investigate the mechanism by which BLM depletion enhances cell survival upon impaired R-loop removal, we assessed BLM recruitment at TC-DSBs upon SETX depletion. ChIP revealed that BLM binding increases at DSBs in SETX-deficient cells (Fig. 4a), further suggesting a role of BLM at TC-DSB sin the absence of R-loop dissolution.

Given the previously reported function of BLM in promoting DNA synthesis at ALT telomeres[51,52] we further set out to measure DNA synthesis at sites of DSBs. For this we implemented a method to directly purify newly synthetized DNA following DSB induction using in vivo EdU incorporation, followed by click chemistry and purification. In control cells, we could readily detect DSB-induced, repair-associated DNA synthesis at the vicinity of DSBs (Supplementary Fig. 5a), compared to replicative synthesis at an origin of replication. In SETX-deficient cells, repair-associated DNA synthesis was increased (Supplementary Fig. 5b, Fig. 4b, c), even though RAD51 filament assembly and hence HR-driven DNA synthesis are impaired in this context (ref. [18], Fig. 3f). Enhanced DNA synthesis in SETX-deficient cells was reduced upon DRB treatment (Supplementary Fig. 5c), suggesting a role of RNA:DNA hybrids. Moreover, repair DNA synthesis upon SETX depletion was not affected by RAD51 depletion (Supplementary Fig. 5d). Importantly, increased DNA synthesis observed in SETX-deficient cells, was partially dependent on BLM (Fig. 4c). Such an effect of BLM depletion on repair associated DNA synthesis was only observed in SETX-deficient cells and not in SETX-proficient cells (Supplementary Fig. 5e), suggesting a specific role of BLM in the absence of R-loop resolution.

In order to investigate DNA synthesis at the genome-wide scale, we further subjected EdU pull-down samples to high throughput sequencing (EdU-seq). EdU-seq was performed before and after DSB induction, in control cells or upon depletion of SETX and/or BLM, and upon prior treatment with mimosin in order to detect only repair associated synthesis (and not DNA synthesis during replication). EdU incorporation revealed that repair synthesis occurred on ~5–10 kb around DSBs in average (Supplementary Fig. 6a) and increased at DSBs produced in RNAPII-enriched loci (TC-DSB) compared to DSBs induced in silent and intergenic loci (Supplementary Fig. 6a, b). In agreement with EdU-qPCR data obtained at a selected AsiSI-induced DSBs (Fig. 4c), averaged EdU-seq profiles around best-induced DSBs showed enhanced DNA synthesis upon SETX depletion, which was further reduced upon BLM co-depletion (Fig. 4d). Notably, upon SETX depletion, both the level and distribution of RNA:DNA hybrids accumulating at DSBs correlated with the repair-associated DNA synthesis detected by EdU-seq (Fig. 4e, Supplementary Fig. 6c). Altogether these data suggest that excessive RNA:DNA hybrid accumulation promotes BLM recruitment at TC-DSBs, which further enhances DNA synthesis in a manner that coincides with RNA:DNA hybrid distribution.

**POLD3-dependent DNA synthesis upon SETX depletion.** In *S. cerevisae*, triple knock-out strains for *rnaseH1*, *rnaseH2*, and *Sen1* (SETX yeast ortholog) are deficient in R-loop removal- and initiate POL32-break-induced replication (BIR) at sites of R-loop-induced damages, with strong consequences on viability[59,60]. Moreover, at ALT telomeres, BLM-dependent DNA synthesis also relies on POLD3, the human POL32 yeast ortholog[51,52,61]. We hence further investigated the consequence of siRNA-mediated POLD3 depletion (Supplementary Fig. 7a) on repair DNA synthesis observed upon excessive RNA:DNA hybrid accumulation. Notably, POLD3 depletion reduced repair DNA synthesis observed in SETX-depleted cells (Fig. 5a), but not in control, SETX-proficient cells (Supplementary Fig. 7b). In addition, similar to BLM, POLD3 depletion rescued DSB-induced defective cell survival observed upon SETX knock-down when compared to SETX-only deficient cells (Fig. 5b, Supplementary Fig. 7c), as did depletion of POLD1, the catalytic subunit of Polδ (Supplementary Fig. 7a, d). Co-depletion of BLM and POLD3 did not significantly further rescue cell survival in SETX-deficient cells suggesting that BLM and POLD3 function in the same pathway (Supplementary Fig. 7e). Enhanced cell survival was also observed when depleting the PIF1 helicase (Supplementary Fig. 7a, Fig. 5c), previously reported to promote cell death upon R-loop induced damage in yeast[59].

We previously reported that SETX depletion increased translocation frequency between TC-DSBs, which could potentially account for increased cell toxicity upon SETX depletion. Notably, both BLM and POLD3 depletion rescued the enhanced translocation rate in SETX-deficient cells (Fig. 5d, Supplementary Fig. 7f).

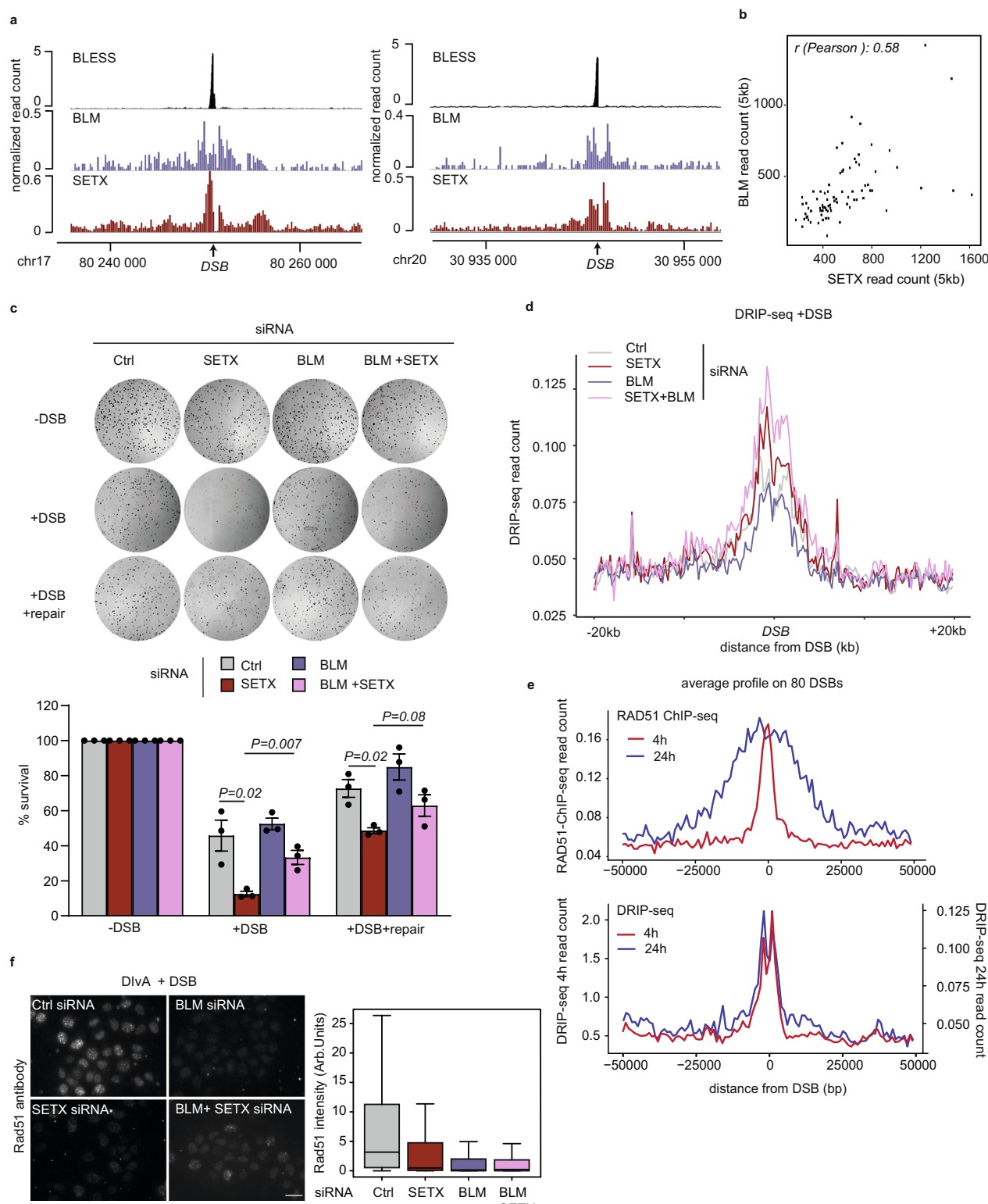

Altogether these data suggest that, upon defective resolution of DSB-induced RNA:DNA hybrids, POLD3- and BLM-dependent DNA synthesis, takes place at DSBs, and contribute to the toxicity triggered by RNA:DNA hybrid accumulation at TC-DSBs.

**Break-induced replication signature relates to BLM and SETX expression levels in pancreatic cancers.** Our above data indicate that POLD3/BLM-dependent DNA synthesis occurs at TC-DSBs

and promotes translocation in SETX-deficient human cells. To further establish whether the interplay between SETX and POLD3/BLM impacts genome stability, we examined genomic signatures in cancer databases with relation to SETX, POLD3 and BLM expression. Since both POLD3 and BLM were previously involved in BIR at telomeres in ALT cells[49,50,52], we focused on the occurrence of copy number alteration, especially tandem duplication (TD) below 100 kb, previously associated to BIR[62]. For this analysis, we used transcriptomic and genomic data

**Fig. 3 BLM deficiency rescues cell lethality in SETX-deficient cells without affecting RNA:DNA hybrids levels. a** Genome browser screenshots representing BLESS, BLM, and SETX ChIP-seq signal obtained after 4 h of DSB induction, at two individual DSBs (chr17:80250841 and chr20: 30946312) (arrows). **b** Scatterplot showing the ChIP-seq BLM and SETX read count read on a 5 kb window around the 80 best-induced DSBs at 4 h after DSB induction. R (Pearson) is indicated. **c** Clonogenic assay in control, BLM, SETX, and BLM/SETX-depleted AID DIvA cells before (−DSB), after (+DSB) DSB induction, and after DSB induction and auxin (IAA) treatment allowing DSB repair (+DSB + repair) as indicated (upper panel). The quantification is represented by the percentage of cell survival in the lower panel. Mean and SEM are shown for $n = 3$ biologically independent experiments. *P*, paired *t*-test (two-sided). **d** Average DRIP-seq profiles obtained post DSB induction (24 h) around the 80 best cleaved DSBs in DIvA cells transfected with Control, SETX, BLM, and SETX + BLM siRNAs. **e** Average RAD51 ChIP-seq (top panel) and DRIP-seq (bottom panel) profiles around 80 best DSBs induced for 4 h and 24 h. **f** (Left) RAD51 immunostaining performed in control, BLM, SETX, BLM/SETX siRNA-depleted DIvA cells after DSB induction for 4 h (DIvA + DSB). Scale bar 20 μM. (Right) RAD51 foci intensity was quantified after DSB induction. siRNA Ctrl, $n = 245$ cells; siRNA BLM, $n = 313$ cells; siRNA SETX, $n = 287$ cells; siRNA SETX + BLM, $n = 304$ cells. Center line: median; Box limits: 1st and 3rd quartile; Whiskers: Maximum and minimum values without outliers. A representative experiment is shown.

available on ICGC database from individual patients with pancreatic cancer (PACA-CA dataset), since it provides a robust set of patient samples in which TD have been annotated. Individual patient genomic data were categorized according to their SETX and POLD3 expression level (i.e., four categories as indicated, Fig. 5e), and the occurrence of TD < 100 kb were compared. Interestingly, in conditions where POLD3 expression is high, TD frequency increases in samples showing low SETX expression compared to samples showing high SETX expression (Fig. 5e, compare red and gray). Importantly, such a SETX expression level dependency is not observed for other genomic signatures such as SNP or INDELs (Supplementary Fig. 7g), excluding a non-specific increase of BIR genomic signature due to an increased load of DNA damage on the genome or an overall decrease of repair capacity in samples expressing low level of SETX. These data suggest that TD signature specifically associates with a R-loop dissolution-deficient background. Reduced expression of POLD3 significantly decreases the frequency of TD < 100 kb in samples that display low expression of SETX (Fig. 5e, compare red and purple), which is in agreement with an involvement of BIR in this increased TD < 100 kb signature. Similarly, reduced BLM expression also decreased TD in patient samples showing low SETX expression (Fig. 5f, compare red and purple). Altogether this further validates our observed genetic interaction between SETX, POLD3, and BLM on cell survival post DSB induction and suggests that the interplay between SETX, POLD3 and BLM directly impacts genome stability in vivo.

## Discussion

Here, we present evidence that the RecQ BLM helicase is a bona fide component of the TC-DSBR machinery and displays a dual role in handling DSBs that occur in actively transcribed loci. We show that BLM preferentially associates with DSBs located in active transcription units, where it fosters resection and RAD51 assembly. Yet in a background where R-loop dissolution is impaired, BLM promotes a toxic, POLD3-dependent, pathway (Fig. 5g).

**Function of BLM in TAHRR, the G2 arm of TC-DSBR**. We found that BLM preferentially associates with DSBs localized in active genes in a manner that depends on transcription (Fig. 1). This behavior is reminiscent of other recently identified TC-DSBR factors, such as SETX[18] or XPG and RAD52[14], whose recruitment at DSBs were also shown to depend on pre-existing transcriptional activity. In agreement with previously reported functions of BLM in HR[28,31–37,63] we found that at these DSBs in active loci, BLM contributes to resection, RAD51 foci assembly and accurate repair (Fig. 2, Supplementary Fig. 3d, e).

However, of interest, depletion of BLM did not reduce cell survival following DSB induction in our system (Fig. 3c). Previous

studies led to conflicting results regarding the sensitivity of BLM-deficient cells to DNA damaging agents. Indeed, both Bloom Syndrome (BS) and BLM-depleted cells were found to be insensitive to hydroxyurea (HU) -induced replication stress, unless prolonged HU treatment is performed[64,65]. Similarly, BLM-defective chicken DT40 cell exhibit normal sensitivity to HU[66]. In contrast, BS and/or BLM siRNA-depleted cells display hypersensitivity to formaldehyde (inducing DNA-protein cross-links[67]), agents inducing DNA interstrand cross-links[68], camptothecin[69], or other genotoxics (for examples see refs. [70,71]). Here, we show that the transient depletion of BLM does not reduce survival of cells that experience a hundred of clean DSBs, a majority of them being induced in active loci[54]. Altogether this suggests that DSBs in transcribed loci mainly utilize HR (TAHRR) to ensure accurate repair, but that TAHRR-deficiency is not cytotoxic even though it affects genome stability. Since our results indicate that BLM mainly functions at TC-DSBs, it is tempting to speculate that the cancer predisposition observed in BS patients arises at least in part from inaccurate repair events at damaged active transcription units, which would hence undergo mutagenic repair while still being proficient for proliferation.

**RNA:DNA hybrid accumulation at TC-DSBs is not strictly related to the availability of single-stranded DNA**. While accumulation of RNA:DNA hybrids at DSBs has been extensively reported in different model systems and using various DNA damaging methods, their mode of accumulation is still unclear[22]. It was suggested that such hybrids may form due to de novo synthetized RNA from DNA ends, by RNAPII or RNAPIII that would hybridize with resected ssDNA and protect it from degradation[23,26,72]. Whether long-range resection promotes RNA:DNA hybrid formation[26] or whether RNA:DNA hybrids promote resection[16,23,73] is currently unclear. Our data suggest that RNA:DNA hybrid accumulation *in cis* to DSBs induced in transcribed loci, which are major sites of hybrid formation post DSB[15,18], is not directly linked to ssDNA availability (Fig. 3). Indeed, BLM depletion, which decreases ssDNA formation, did not reduce RNA:DNA hybrid levels *in cis* to DSB. In addition, RNA:DNA hybrid profiles did not follow the RAD51 pattern upon prolonged time of DSB induction, with RAD51 spreading on ~30–40 kb around DSB while RNA:DNA hybrids were only detected on a limited region (<10 kb around DSB) independently of the duration of break induction. Hence, our data argue against a de novo transcribed lncRNA that would hybridize with the entirety of the ssDNA[26] and rather support the hypothesis that RNA:DNA hybrids, and hence rather R-loops (three-stranded nucleic acid structures composed of an RNA:DNA hybrid and the displaced DNA strand) actually form at TC-DSB, prior to resection[23].

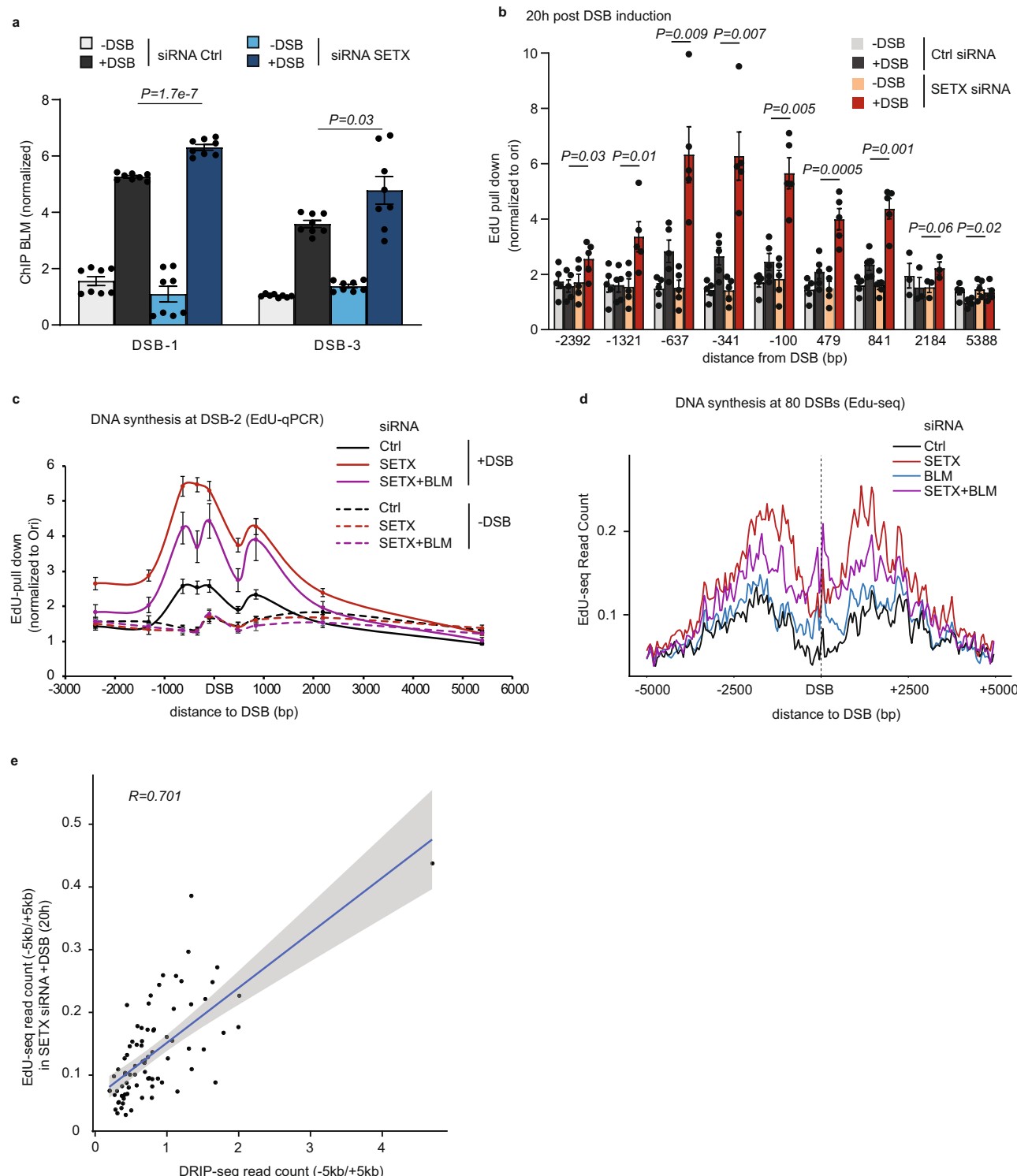

**The toxicity of RNA:DNA hybrids at TC-DSBs is not linked to TAHRR deficiency but to POLD3/BLM-dependent DNA synthesis.** SETX deficiency strongly impairs cell survival following DSBs in active loci[18]. Given that once damaged, transcribed loci accumulate RNA:DNA hybrids and that their dissolution is mandatory for RAD51 nucleofilament formation and proper execution of HR[14,18], we previously postulated that HR deficiency may account for the observed decreased cell survival in SETX-deficient cells[18]. Here we found that inefficient RAD51 loading is not responsible for DSB-induced lethality upon impaired R-loop

removal. BLM depletion was indeed able to partially rescue SETX-induced lethality while not being able to restore RAD51 foci formation. Hence the toxicity of sustained DSB-induced RNA:DNA hybrids is not due to a decreased HR capacity.

Rather we found that BLM and POLD3-dependent DNA synthesis is responsible for increased cell death induced by TC-DSBs when R-loop resolution is impaired. Indeed (i) depletion of POLD3/POLD1 and BLM rescues lethality upon DSBs in active genes in a SETX-depleted background, (ii) excessive DNA synthesis takes place at DSBs upon SETX depletion despite

**Fig. 4 BLM promotes DNA synthesis *in cis* to TC-DSB in a SETX-deficient background. a** BLM ChIP performed in DIvA cells transfected with control or SETX siRNA before and after DSB induction (4 h) and analyzed by qPCR at two DSBs (DSB-1 chr22:38864101 and DSB-3 chr20:30946312). Data are normalized to a control location devoid of DSB. Mean and SEM are shown for $n = 8$ technical replicates of a representative experiment. P, unpaired *t*-test (two-sided). **b** Repair synthesis measured by EdU-pull down on a $-/+ 3$ kb region around the DBS-2 (chr9:130693170) in control and SETX siRNA-depleted, before and after DSB induction as indicated (20 h DSB induction). Data are shown as values normalized to the Ori region. Mean and SEM are shown for $n = 5$ biologically independent experiments. P, paired *t*-test (two-sided), between Ctrl siRNA + DSB and SETX siRNA + DSB. **c** EdU-pull down efficiency in control, SETX and SETX/BLM siRNA depleted DIvA cells before (−DSB) and after DSB induction for 20 h (+DSB), at a DSB bound by BLM (DSB-2, chr9:130693170). Data are normalized to an origin of replication (Ori). Mean and SEM of $n = 4$ biologically independent experiments are shown. **d** Average EdU-seq profile around the 80 best-induced DSBs after DSB induction (20 h), in DIvA cells treated with mimosine (24 h) and transfected with control, BLM, SETX, or BLM + SETX siRNA as indicated. **e** Scatterplot showing the level of EdU-seq signal ($-5/+ 5$ kb) and DRIP-seq ($-5/+5$ kb) obtained in SETX-deficient cells at each DSB ($n = 80$). R (Pearson) is indicated. Error in gray represents the regression confidence as obtained by the geom_smooth function of ggplot2 (linear model).

reduced RAD51 loading, (iii) this repair-associated DNA synthesis depends on POLD3 and BLM and (iv) DNA synthesis level and profile coincide with RNA:DNA hybrid distribution. Similarly, previous studies in yeast showed that DSBs in strains accumulating R-loops (Δ *rnaseh1*, *rnaseh2*) displayed delayed repair, eliciting as strong decrease in viability which also depends on a POL32-driven BIR pathway[59]. How BIR triggers cell death in this context deserves further investigation, since it was shown to not be triggered by checkpoint activation in yeast[59,60]. We can postulate that cell death may be triggered by a high genomic instability, as suggested by the increased rate of translocation in SETX-deficient cells.

Of interest, yeast BIR requires SGS1 (the BLM ortholog) and is fostered by the deletion of the helicase MPH1 (the FANCM ortholog)[45], which displays RNA:DNA hybrid unwinding activity[74]. In addition, in human cells, a POLD3-dependent pathway is involved in Alternative Lengthening of Telomeres (ALT)[46,47], in a manner that not only depends on BLM[49,50,52] but also increases upon accumulation of R-loops[48,75–77]. Since DIvA cells were established in U2OS ALT cells, whether the POLD3 pathway at TC-DSB described here only takes place in ALT cells deserves further investigations. Interestingly, in addition to alternative telomere lengthening, POLD3-dependent DNA synthesis also occurs at one-ended DSBs arising from collapsed replication forks or at fragile genomic locations, during mitosis (known as MiDAS, Mitotic Induced DNA synthesis)[62,78–81], and has been associated with genomic instability[62]. Notably evidence suggests that both ALT DNA synthesis and MiDAS at common fragile sites are RAD51-independent[52,62,78], as is the mechanism described in this study at intra-chromosomal TC-DSB in the absence of R-loop removal (Supplementary Fig. 5). Moreover, as observed for yeast and human BIR[81,82], the pathway that operates at TC-DSB is PIF1-dependent (Fig. 5).

Altogether, our data suggest that a similar POLD3/BLM/PIF1-dependent and RAD51-independent pathway can operate at intra-chromosomal two-ended DSBs in human cells, especially to handle TC-DSBs upon excessive R-loop accumulation. While awaiting further studies, we speculate that the R-loop accumulation that takes place at DSBs induced in active loci may actually convert a two-ended DSB into a "one-ended-like" DSB. This, for instance, may occur upon asymmetrical RNA:DNA hybrid accumulation making one end more difficult to process. Such an hypothesis would be in agreement with a recent study showing that impaired synchronous resection at the two ends triggers BIR at two-ended breaks in yeast[83].

Of interest, during ALT, telomeres clustering favors BIR[49,51,84]. Interestingly, we previously reported that TC-DSBs display clustering[13]. Whether the POLD3-dependent pathway described here, which handles TC-DSB upon unscheduled R-loop accumulation, is also potentiated by their ability to cluster thus deserves further investigations.

In conclusion, our work indicates that during DSB repair, RNA:DNA hybrids are highly toxic intermediates that potentiate the use of a BLM/POLD3-dependent pathway which trigger cell death. In this context, chemical compounds stabilizing R-loops in cancer backgrounds where the BIR pathway is intact may reveal a good strategy to enhance the potency of gene-specific genotoxic chemotherapeutic drugs such as Topoisomerase II poisons.

## Methods

**Cell culture**. DIvA (AsiSI-ER-U2OS) and AID-DIvA (AID-AsiSI-ER-U2OS) cells were cultured in Dulbecco's modified Eagle's medium (DMEM) supplemented with antibiotics, 10% FCS (Invitrogen) and either 1 μg/mL puromycin (DIvA cells) or 800 μg/mL G418 (AID-DIvA cells) at 37 °C under a humidified atmosphere with 5% $CO_2$. For AsiSI-dependent DSB induction, cells were treated with 300 nM 4 hydroxytamoxifen (4OHT, Sigma; H7904) for 4 h or 20 h as indicated. When indicated, 4OHT treated cells were washed 3 times with pre-warmed PBS and further incubated with 500 μg/mL auxin (Sigma; I5148). For transcriptional inhibition, DRB (100 μM) was added to the medium 1 h prior to 4OHT (4 h) treatment. To inhibit replicative DNA synthesis, cells were treated with 400 μM mimosine for 24 h prior cell collection. siRNA transfection was performed 48 h prior 4 h 4OHT treatment. siRNA sequences are detailed in Supplementary Table 1. For cell synchronization (Supplementary Fig. 2b), double thymidine block was performed as previously described[55]. Briefly, cells were treated for 18 h with 2 mM thymidine, released for 12 h and treated again with thymidine for another 18 h. Cells in G1 or in G2 phase of the cell cycle were, respectively, collected 15 h or 6 h after the second release. DSBs were induced 4 h prior harvesting the cells.

**Clonogenic assays**. After siRNA transfection, 4000 AID-DIvA cells were seeded in 10 cm diameter dishes. Forty-eight hours later cells were treated with 300 nM 4OHT for 4 h and, when indicated, washed three times with pre-warmed PBS and further incubated with 500 μg/mL auxin for another 4 h. After three washes with pre-warmed PBS, complete medium was added to each dish. After 7 to 10 days, cells were stained with crystal violet (Sigma) and counted. Only colonies containing more than 50 cells were scored.

**Cell cycle analysis**. Cells were washed with PBS, fixed with pure ethanol for at least 30 min at 4 °C and washed with PBS-Tween 0.5%. Finally, cells were resuspended in staining solution (0.2 mg/mL RNase A, 50 μg/mL propidium iodide) and incubated for 1 h at 37 °C prior FACS analysis using a Cytoflex S (Beckman Coulter) flow cytometer.

**Chromatin immunoprecipitation followed by qPCR and high throughput sequencing**. ChIP assays were carried out according to the protocol described in ref. [54]. Briefly, after crosslinking with formaldehyde (1%, 15 min at room temperature), glycine (0.125 M) was added for 5 min. Cells were washed twice with cold PBS, harvested by scraping, and further incubated in lysis buffer (50 mM Pipes pH 8, 85 mM KCl, 0.5%NP-40) and homogenized with a Dounce homogenizer. Nuclei were further incubated in nuclear lysis buffer: (50 mM Tris pH 8.1, 10 mM EDTA, 1%SDS) and sonicated (10 x) for 10 s at a power setting of 5 and 50% duty cycle (Branson Sonifier 250). Samples were diluted ten times in dilution buffer (0.01% SDS, 1.1% Triton X-100, 1.2 mM EDTA, 16.7 mM Tris pH, 8.167 mM NaCl) and precleared for 2 h with 100 μL of protein-A and protein-G beads (Sigma), previously blocked with 500 μg of BSA 2 h at 4 °C. 200 μg of precleared chromatin was immunoprecipitated overnight at 4 °C by using 2 μg of anti-BLM (Abcam, ab2179), anti-XRCC4 (Abcam, ab145), anti-RAD51 (Santa Cruz, SC-8349), anti PARP1 (Cell Signaling, 9542), anti-Histone H3 (Abcam, ab1791) or without antibody (mock). 100 μL of blocked protein A/protein G beads were further added for 2 h at 4 C on a rotating wheel. Immunoprecipitated samples were washed with dialysis buffer (2 mM EDTA, 50 mM Tris pH 8.1, 0.2% Sarkosyl), with

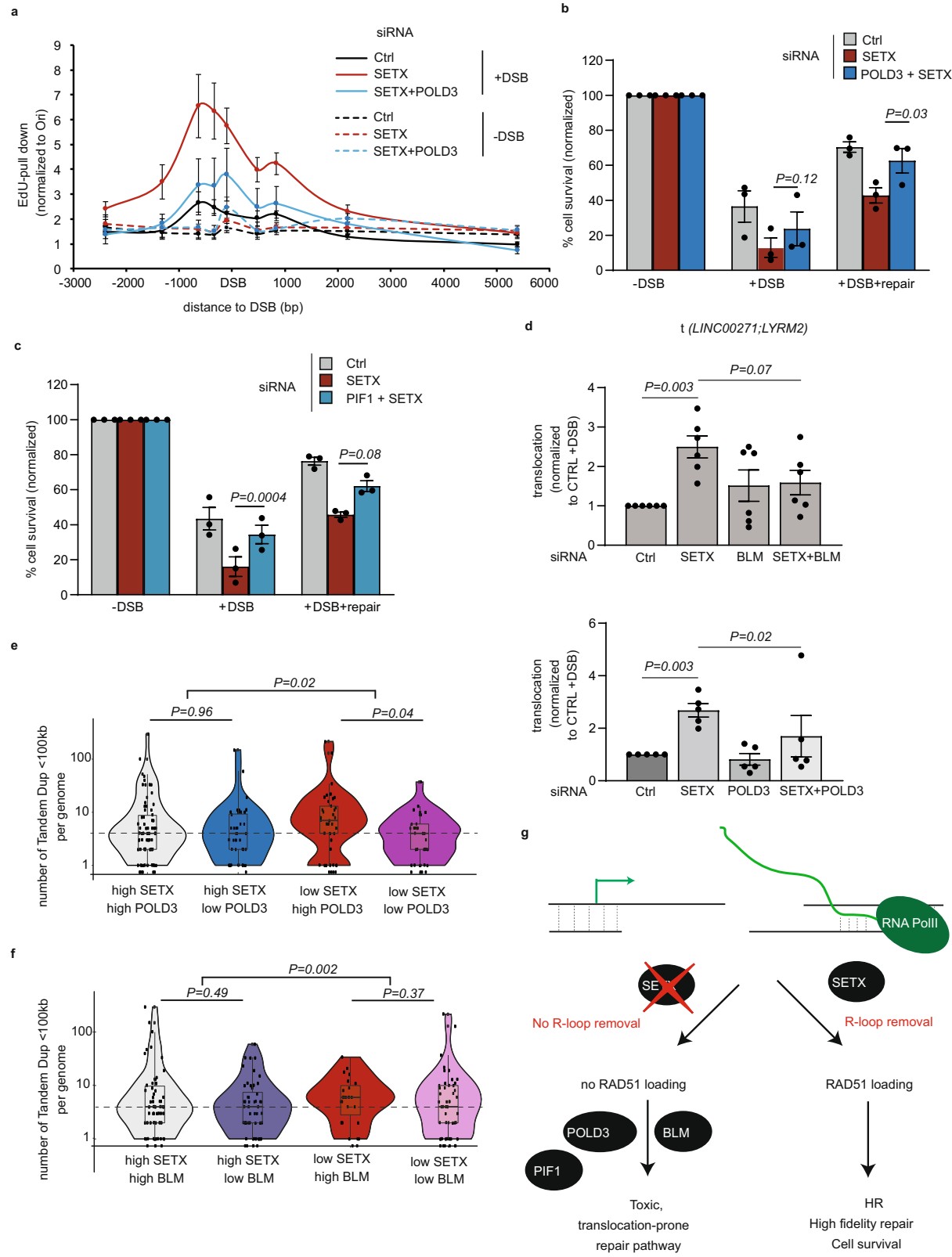

wash buffer (100 mM Tris pH 8.8, 500 mM LiCl, 1%NP-40, 1%NaDoc) (4 times) and re-suspended in 200 µL of TE buffer (10 mM Tris pH 8, 0.5 mM EDTA pH 8) with 30 µg of RNase A for 30 min at 37 °C. Crosslink reversal was performed in the presence of 0.5% SDS at 70 °C overnight with shaking. After 2 h proteinase K treatment, immunoprecipitated and input DNA were purified with phenol/chloroform, precipitated, and dissolved in 100 µL water. For quantitative PCR analysis, both input and IP samples were analyzed using the primers described in Supplementary Table 2. IP efficiencies were calculated as the percent of input DNA

immunoprecipitated. Quantitative PCR was performed in a CFX384 real time system (Bio-Rad) using Bio-Rad CFX Manager version 3.1 software.

**DRIP-seq.** DRIP-seq was carried out according to the protocol described in ref. [18] on DIvA cells, treated with 300 nM 4OHT for 4 h or 24 h as indicated, and using 10 µL of S9.6 antibody (1 mg/mL, kind gift from P. Pasero, IGH, France). Samples from 3 independent DRIP experiments were pooled and sonicated to an average

**Fig. 5 POLD3 contributes to DNA synthesis at TC-DSBs in SETX-depleted cells. a** EdU-pull down efficiency in control, SETX and SETX/POLD3 siRNA depleted DIvA cells before (-DSB) and after DSB induction for 20 h (+DSB), at a TC-DSB bound by BLM (DSB-2, chr9:130693170). Data are normalized to an origin of replication (Ori). Mean and SEM of $n = 4$ biologically independent experiments are shown. **b**, **c** Percentage of cell survival from clonogenic assay in AID-DIvA cells transfected with siRNA (as indicated) before (−DSB), after (+DSB) DSB induction (4 h), and after repair (+DSB + repair). Mean and SEM of $n = 3$ biologically independent experiments are shown. P, paired t-test (two-sided). **d** Translocation frequency t(*LINC0072;LYRM2*) analyzed by qPCR in AID DIvA cells transfected with siRNA as indicated, after DSB induction and repair. Values are plotted as mean +/− SEM of $n = 6$ (top panel) and $n = 5$ (bottom panel) of biologically independent experiments. P, paired t-test (two-sided). **e**, **f** Violin Plots representing the number of tandem duplication (<100 kb) per genome depending on the indicated gene expression using pancreatic cancer gene expression and genomic data available on ICGC database (PACA-CA project). P values are indicated (non-parametric two-sided Wilcoxon's test). P value assessing the significance between ratios (whether "High SETX" group behaves differently than "Low SETX" group) is also indicated. **e**, from left to right, $n = 76$; $n = 41$; $n = 41$; $n = 37$; **f**, from left to right, $n = 61$; $n = 56$; $n = 25$; $n = 53$. Center line: median; Box limits: 1st and 3rd quartile; Whiskers: Maximum and minimum values without outliers. **g** Schematic representation of the function of SETX and BLM in TC-DSBR. In WT conditions, SETX removes R-loops that accumulate at DSBs induced in transcribed loci (TC-DSB), thereby allowing RAD51 loading, faithful DSB repair by HR and cell viability. In SETX-depleted cells, deficient R-loop removal reduces RAD51 loading and HR repair. DSBs are further handled by a BLM/POLD3-dependent, RAD51-independent pathway, triggering an increase of tandem duplications on the genome and impairing cell survival.

size of 300 bp using a Bioruptor (Diagenode) for 20 cycles of 30 s On, 30 s Off, High setting. Immunoprecipitated DNA was subjected to library preparation and single-end sequencing on a NextSeq 500 at EMBL GeneCore (Heidelberg, Germany).

**Site-specific DNA synthesis quantification by EdU-qPCR and EdU-seq**. After siRNA transfection by electroporation $4 \times 10^6$ cells were seeded in 15 cm diameter dishes. Forty-eight hours after, cells were treated with 400 μM mimosine 24 h prior harvesting when indicated (for EdU-Seq), and 300 nM 4OHT (Sigma; H7904) for 4 h or 20 h as indicated. EdU (5-ethynyl-2'-deoxyuridine from Invitrogen; 10 μM final) was added 1 h before cells were harvested. Genomic DNA was extracted with TEN/RNAse buffer (10 mM Tris pH 8, 100 mM NaCl, 25 mM EDTA, 0.5% SDS, 20 μg/mL RNAse A) for 1 h at 37 °C before adding 20 μL of Proteinase K (10 mg/mL, Roche) for 2 h at 56 °C. Another 20 μL of Proteinase K was added for O/N digestion. Genomic DNA was then purified with Phenol/chloroform (Sigma), precipitated with 70% ethanol (Sigma)/Na Acetate (0.3 M, Invitrogen), and digested with MboI (125U per 100 μL reaction volume) for 12 h at 37 °C followed by NlaIII (12.5U per 100 μL reaction volume) for 4 h at 37 °C. Digested DNA was purified with Phenol/chloroform and precipitated as above. DNA was then biotinylated by click-it reaction according to the manufacturer protocol (Click-iT® Nascent RNA Capture Kit; Life Technology) and precipitated with 70% EtOH (Sigma)/Na-acetate (0.3 M, Invitrogen). Input DNA was taken at this step. Biotinylated DNA was pull-down using streptavidin beads according to manufacturer protocol (Click-iT® Nascent RNA Capture Kit; Life Technology). Pull-down and input DNA were analyzed by qPCR using the primers indicated in Supplementary Table 2. For EdU-seq, DNA concentration on beads was measured using Qubit quantification and DNA on beads was resuspended in 50 μL Elution buffer from Arima Hi-C Kit (BoxA). The End Repair, dA-tailing, and Adapter ligation steps were done using the NEB Next Ultra II DNA library Prep kit for Illumina as suggested by the manufacturer (Arima Hi-C Kit User guide).

**Sequencing data analyses**. For BLM, RAD51, and SETX ChIP-Seq after 4 h and 24 h 4OHT treatment, sequencing libraries were prepared by using 10 ng of purified DNA (averaged size 250–300 bp) and subjected to high throughput sequencing (single-end) using a HiSeq 2000 sequencing (BGI, Hong-Kong) or using Illumina NextSeq 500 (EMBL Genomics core facilities Heidelberg, Germany). ChIP-Seq experiments were aligned using bwa (0.7.12-r1039) on human reference genome hg19. Samtools 1.9 was used for sorting and indexing, and a custom R script with R package rtracklayer in R 3.6.3 software was used to generate normalized (CPM) coverage of mapped reads in bigWig format. ChIP-Seq average profiles were computed using normalized number of reads around the 80 best cleaved DSBs[55]. Data were visualized using Integrated Genome Browser version 9.1.6. For heatmaps representation (performed using Java Treeview (http://www.jtreeview.sourceforge.net)) each tile shows the average normalized number of reads at each genomic location centered around DSBs, ordered based on the BLESS signal (Fig. 1b). For boxplots, the sum of the normalized number of reads of ChIP-Seq was computed on a given window around each DSB (as indicated). For comparison with RNAPII ChIP-seq data (Fig. 1d), each RNAPII categories were computed using coverage of normalized reads counts (ChIP-seq data in DIvA cells from ref. [18]) from a given windows of 10 kb around each DSBs and divided in four groups (of 20 DSBs each). Box-plots were generated with R-base (tidyverse 1.3.0, including ggplot2 R package 3.3.3). The center line represents the median, box ends represent, respectively, the first and third quartiles, and whiskers represent the minimum and maximum values without outliers. Outliers were defined as first quartile − (1.5 × interquartile range) and above third quartile + (1.5 × interquartile range). Statistical hypothesis testing was performed using non-parametric paired Mann–Whitney–Wilcoxon (wilcoxon.test() function in R) to test distribution differences between two populations. To determine categories of DSBs enriched in

BLM or not, for each cleaved DSBs, BLESS[55] and BLM normalized coverage was computed using similar method used for boxplots, using a 1 kb and 4 kb window, respectively. Then the ratio between these two values was computed and used to sort DSBs. The first 20 DSBs were considered as BLM-high and the last 20 as BLM-low.

DRIP-Seq samples were sequenced using Illumina NextSeq 500 (single-end 85 bp reads) at EMBL Genomics core facilities (Heidelberg, Germany). The quality of each raw sequencing file (fastq) was verified with FastQC, reads were aligned to the reference human genome (hg19) using bwa (http://bio-bwa.sourceforge.net/) and further processed using samtools (http://www.htslib.org/) for duplicate removal (rmdup), sorting (sort) and indexing (index). Coverage for each aligned ChIP-seq dataset (.bam) was computed with the rtracklayer R package and normalized using total read count for each sample. Coverage data was exported as bigwig (file format). Averaged DRIP-Seq profiles were generated using the R package ggplot2.

For EdU-seq, the quality of each raw sequencing file (fastq) was verified with FastQC, fastq files were aligned using bwa on human reference genome hg19. Samtools was used for sorting and indexing, and a custom R script with R package rtracklayer was used to generate normalized (CPM) coverage of mapped reads in bigWig format.

**Repair kinetics and repair fidelity at AsiSI sites**. Repair kinetics at specific AsiSI-induced DSBs were measured as described in ref. [12] by a cleavage assay permitting the capture of unrepaired DSBs, at the indicated times after auxin addition. For repair fidelity assays, AID-DIvA cells were treated with 4OHT to induce DSBs for 4 h (300 nM) followed by an auxin treatment for 4 h. The next day, cells were treated again with 4OHT for 4 h. DNA was extracted and subjected to a cleavage assay as described in ref. [58]. Primers used are detailed in Supplementary Table 2.

**Immunofluorescence**. Immunofluorescence against γH2AX (Sigma-Aldrich, JBW301), and RAD51 (Santa Cruz, SC-8349) in AID-DIvA cells was performed as in ref. [14]. Image acquisition was performed using MetaMorph version 7.1.0.0 on a wide-field microscope equipped with a cooled charge-coupled device camera (CoolSNAP HQ2), using a ×40 objective (for quantification). Quantification was performed using Columbus, an integrated software to the Operetta automated high-content screening microscope (PerkinElmer). DAPI stained nuclei were selected according to the B method and appropriate parameters, such as the size and intensity of fluorescent objects, were applied to eliminate false-positive. Then γ-H2AX or RAD51 foci were detected with the D method with the following parameters: detection sensitivity, 1; splitting coefficient, 1; background correction, >0.5 to 0.9.

**Laser-induced DNA damage and real-time recruitment of BLM to DSBs**. For real-time recruitment of BLM, $1 \times 10^6$ U2OS Cells (ATCC) were seeded onto No. 1.5 Gridded glass bottomed 35 mm dishes (MatTek) and transfected with pcDNA3.1_GFP-BLM plasmid using JetPrime (Polyplus-transfection). Transfected cells were grown in 10% fetal bovine serum (Sigma-Aldrich) containing Dulbecco's modified Eagle's medium (Life Technologies) under standard conditions. 24 h post-transfection, DSBs in GFP-BLM expressing cells were generated using a Stanford Research Systems (SRS) NL100 nitrogen MicroPoint system (Photonics Instruments) equipped to a Nikon Eclipse TE2000 spinning disk confocal microscope (Nikon Instruments Inc.). The microscope was supported with temperature and CO2-regulated incubation chamber. Site-specific DSBs in 0.25 × 3 μM tracks were induced with 435 nm laser, regulated through Volocity software 6.3 (Perkin-Elmer). Following irradiation, the recruitment of GFP-tagged BLM was recorded at 15 s interval for 5 min with a CCD camera (Hamamatsu). GFP signal intensity in the DSB-induced laser tracks was quantified and normalized to pre-DSB signals

using Velocity imaging software. The results are presented as mean +/− SEM (Standard Error of Mean).

Cell cycle status of the laser-irradiated cells was determined by immunostaining for cyclin A2. Briefly, irradiated cells were washed with PBS, fixed (3.7% formaldehyde in PBS for 10 min), permeabilized (0.1% Triton X-100 in PBS for 10 min) and blocked with 5% FBS (1 h at room temperature). G1 and S/G2 phase cells were detected by immunostaining the cells with mouse anti-cyclin A2 antibody (1:3000, ab16726, Abcam) for 1 h followed by donkey anti-mouse Alexa 594 antibody (1:1000, A-21203, Life Technologies) for 1 h at 37 °C. Cells were washed four time with 0.1% Tween-20 in PBS for 40 min after incubation with antibodies, and mounted in ProLong Gold antifade mounting media with DAPI (Life Technologies). Laser-irradiated cells from gridded plates were traced and imaged with a confocal microscope and analyzed with Velocity software.

**Resection assay**. Measure of resection was performed as described in ref. [57] with the following modifications. DNA was extracted from fresh cells using the DNAeasy kit (Qiagen) and directly digested using the Ban I restriction enzyme that cuts at ~200 bp and 950 bp from a DSB located on chromosome 22 and enriched in BLM (chr22:38864101). Digested and undigested DNA were analyzed by qPCR using the primers described in Supplementary Table 2.

**ICGC cancer data analysis**. Publicly available cancer data from ICGC consortium data portal release 28 was retrieved (https://dcc.icgc.org/), with a focus on PACA-CA cohort (pancreatic cancer) for which tandem duplication mutations were available in enough patients for statistical analyses. For each patient of the cohort, the number of somatic tandem duplications whose sizes were shorter than 100 kb (from structural somatic mutation data) was computed. The patients were next divided into 4 groups depending on the expression SETX and either POLD3 or BLM: patients with high SETX expression and high POLD3 or BLM expression, patients with high gene SETX expression and low POLD3 or BLM expression, patients with low SETX expression and high POLD3 or BLM expression, and patients with low SETX expression and low POLD3 or BLM expression. The number of tandem duplications between two groups was compared by a Wilcoxon's test (unpaired). To compute the effect of one gene on the number of tandem duplications depending on the other gene, we used a negative binomial regression with an interaction term between the two genes and computed the corresponding p-value.

**Western blot**. Western blot analysis was performed as previously described[18]. Anti-BLM antibody (Abcam, ab2179) and anti-tubulin (Sigma, T6557) were used at 1:1000 and 1:10,000, respectively. Chemiluminescence was visualized using a ChemiDoc™ Touch Imaging System (Bio-Rad) using Image Lab Touch version 1.2.0.12 software.

**Reporting summary**. Further information on research design is available in the Nature Research Reporting Summary linked to this article.

## Data availability
The data that support this study are available from the corresponding author upon reasonable request. All high throughput sequencing data generated in this study are available in the ArrayExpress database (http://www.ebi.ac.uk/arrayexpress) under accession number E-MTAB-11592 (BLM and RAD51 (after 24 h of DSB induction) ChIP-seq, EdU-seq, and DRIP-seq data). Information related to previously generated datasets can be found in the respective publications: total RNAPII, RNAPII-S2P, RAD51 (after 4 h of DSB induction), and XRCC4 ChIP-seq[12]; γH2AX ChIP-seq[54]; BLESS[55] and SETX ChIP-seq[18].

## Code availability
Customs codes used in this study have been deposited on GitHub, https://github.com/LegubeDNAREPAIR.

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

## Acknowledgements

Sample sequencing were performed by the EMBL Genomic Core, and the ICGex NGS platform of the Institut Curie, supported by the grants from Agence Nationale pour la Recherche, "Investissements d'Avenir" program (ANR-10-EQPX-03 and ANR-10-INBS-09-08 (France Génomique Consortium)), by the Canceropole Ile-de-France and by the SiRIC-Curie program-SiRIC Grant "INCa-DGOS- 4654". This work was initiated by a CEFIPRA collaborative grant (4603-1). Funding in GL laboratory was provided by grants from the European Research Council (ERC-2014-CoG 647344; ERC-2021-AdG-101019963), Agence Nationale pour la Recherche (ANR-14-CE10-0002-01; ANR-18-CE12-0015; ANR-21-CE12-0033-03), Fondation Bettencourt-Schueller (Coup d'Elan) and the Ligue Nationale contre le Cancer (LNCC). S.C. and I.L. were supported by a fellowship from the Fondation pour la recherche medicale (FRM) and Fondation ARC, respectively. V.A.B. lab was funded by the Intramural Program of the National Institute on Aging, NIH, USA. V.B. laboratory was funded by grants from Agence Nationale pour la Recherche (ANR-15-CE11-0011, ANR-18-CE12-0018 and ANR-21-CE12-0033-03), from Electricité de France, and from Fondation pour la Recherche Médicale (Equipe FRM EQU201903007785). T.C. and N.P. are INSERM researchers.

## Author contributions

I.L., S.C., A.G., T.C., A.M., C.A, N.F., and N.P. performed and analyzed experiments. M.A., V.R., R.M., and M.G. performed bioinformatic analyses of ChIP-seq, Edu-seq, and DRIP-seq datasets and ICGC database. S.A.R. performed GFP-BLM recruitment assays under the supervision of V.A.B. D.V.V. optimized EdU Pull Down under the supervision of V.B. A.M., T.C., N.P., and G.L. conceived and supervised experiments. G.L. wrote the manuscript. All authors commented and edited the manuscript.

## Competing interests

The authors declare no competing interests.
