## [Peer Review File · Nature Communications]

REVIEWER COMMENTS

Reviewer #1 (Remarks to the Author):

In this manuscript, the authors show that BLM is recruited to DSBs at sites of active transcription. Consistent with the known functions of BLM, depletion of BLM reduces DNA end resection, RAD51 recruitment, and DSB repair accuracy. Because SETX and BLM are similarly recruited to DSBs in active genes, the authors investigated their functional relationship. Interestingly, depletion of BLM rescued the viability of SETX knockdown cells in the presence of DSBs. This rescue by BLM is not attributed to a reduction in resection or R-loops, but correlated with a reduction in EdU incorporation at DSBs. Depletion of POLD3 and PIF1 also reduced EdU incorporation at DSBs and rescued the viability of SETX knockdown cells after DSB induction, which suggest that BIR is responsible for the toxic DNA synthesis. The authors proposed an interesting model that BLM suppresses the induction of toxic BIR by R-loops. Although the model of this is interesting, the current data are still underdeveloped. Addressing the comments would help strengthen the proposed model.

1. How is BLM preferentially recruited to DSBs in active genes? How does BLM spread 5-10 kb around DSBs? Some mechanistic details would be helpful.
2. Given that BLM is required for efficient DNA end resection and RAD51 recruitment (Fig. S2C, S2D), it is surprising that BLM knockdown did not affect DSB repair efficiency (Fig. S2F). An explanation is needed.
3. In Fig. 2C, is the helicase activity of BLM required for the reduction of viability in siSETX cells?
4. In Fig. 2D, although siBLM cells are defective in resection, siSETX+siBLM cells have more R-loops than siSETX cells. Why would siSETX increase R-loops more in siBLM cells? Should one expect that siBLM reduces R-loop levels even in the absence of SETX?
5. In Fig. 3A, can the authors test if the EdU incorporation at DSBs is RAD51 and/or R-loop dependent?
6. In Fig. 3, are BLM and POLD3 'epistatic' for the EdU incorporation in siSETX cells?
7. Although the EdU incorporation at DSBs in siSETX cells correlates with the reduction in cell survival, there is no direct evidence that this DNA synthesis is responsible for the reduction of cell viability. Although disruption of BIR reduces EdU incorporation at DSBs and rescues the viability of siSETX cells, these two effects may not be causally linked. This is an important issue to address. Any evidence that BIR generates toxic DNA structures or secondary DNA damage?
8. The data in Fig. 4 seem underdeveloped. What is the basis to think that BIR is the primary cause of TD <100kb? Although the paper by Costantino et al. suggested that BIR can induce TD, it may or may not be the primary cause of TD in cancer cells. Also, the functional meaning of high or low levels of SETX, POLD3, and BLM is unclear. Does high POLD3 or BLM mean more BIR activity? Are POLD3 and BLM overexpressed in tumors or simply not down regulated? When SETX is low, is it low enough to increase

R-loops? It seems to me that the interpretations of the data in Fig. 4 are based on a number of untested presumptions. It will take more work to substantiate these interpretations.

Reviewer #2 (Remarks to the Author):

This is a review of NCOMMS-21-14809. In this study, the authors examine the action of the BLM helicase at chromosomal double strand breaks (DSBs) induced by the nuclease AsiSI. Data are presented that BLM is recruited to a greater degree to DSBs in transcribed loci, the R-loop dissolution factor SETX is important to suppress nascent DNA synthesis at DSBs (EdU labeling / IP method), and that such nascent DNA is promoted by BLM and the Polymerase delta subunit POLD3. The reduction in EdU labeling in the BLM+SETX co-depleted cells vs. SETX depleted alone correlates with partial rescue of viability in the co-depleted vs. SETX depleted alone. Hence, the authors conclude that BLM-mediated DNA synthesis in SETX-depleted cells is toxic. Controls showing these effects of BLM on lethality in SETX-deficient cells are independent of end resection and RAD51-loading are shown. Altogether, the study indicates that DSBs with R-loops are prone to DNA synthesis that is consistent with a break-induced replication mechanism, as found in ALT telomere maintenance. A lot of the data are convincing, but throughout there are many missing controls, and some data that is not as convincing. There are also some issues with interpretation.

Major points: I found the most novel/compelling part of the study to be the EdU labeling at DSBs, but this seems somewhat underdeveloped, as follows.

1. The effect of BLM and POLD3 depletion on EdU labeling is not shown (Fig 3A/B). Presumably, in the presence of SETX, EdU labeling is weak, and as such any role in BLM and POLD3 in promoting these events may not be detectable. However, this should be directly shown, and furthermore the possibility that BLM and POLD3 suppress EdU labeling (in SETX+ cells) should be formally ruled out with this experiment.
2. There is clearly much more to learn about the EdU incorporation at DSBs caused by SETX depletion (what is the template? why are R-loops causing such an induction of EdU labeling? what other transcription / repair defects trigger such EdU incorporation?), and certainly more that can fit in the first report, but two comparisons relevant to this study seem in order a) is the synthesis in SETX-depleted cells affected by RAD51-depletion? b) An assumption is that SETX is acting in cis at these DSBs to clear R-loops and thereby suppress EdU labeling. This could be assessed through at least one of these control experiments: does DRB treatment also cause loss of the EdU-labeling? does RNaseH expression cause a loss in the EdU-labeling?

Other points:

3. The degree to which BLM recruitment to DSBs is affected by transcription level at the locus should be described more rigorously. Namely, while there appears to be a pattern favoring BLM recruitment to transcribed loci, the affect seems relatively mild, particularly in comparison to how RAD51 is recruited. A more rigorous description, which appears to include evidence for BLM recruitment to DSBs in non-

transcribed loci too, will improve the lasting impact of the study.

4. Along the lines of point 1, Figure 1E doesn't have statistical analysis.

5. A key part of their model is that SETX loss should probably cause elevated BLM recruitment, but this is not tested. Even the negative result would be informative and important for

6. As Nature Comms is a long-formal journal, please move many of the Supplemental Figure panels to main figures, particularly Fig S4, if not more of them.

Reviewer #3 (Remarks to the Author):

Cohen et al map the BLM helicase to the vicinity (5-10 kb) of DSBs using their well characterised DivA system. DSBs at actively transcribing genes were particularly enriched in BLM and this enrichment was reduced upon inhibition of transcription. Mechanistically, they demonstrate that BLM is required for resection of DSBs. Previously, these authors have shown similar localisation of SETX, which can resolve R-loops, to DSBs at actively transcribed loci. Surprisingly, BLM depletion partially rescued cell death observed upon depletion of SETX due to defective removal of DNA:RNA hybrids. The authors speculate that as BLM promotes long-range resection, its depletion may rescue DSB-induced cell death in SETX depleted cells by reducing resection-dependent DNA:RNA hybrid formation. Surprisingly, measurement of R-loop formation by DRIP-seq did not reveal significant difference in their formation upon BLM depletion, suggesting that R-loop formation is independent of resection (see major point 1, below).

Unlike BLM, depletion of CtIP, which is required for initiating resection, did not rescue the lethality observed upon DSB induction in SETX-depleted cells. The authors have interpreted this as evidence for BLM promoting a cytotoxic pathway upon the excessive R-loop formation upon SETX depletion. Measurement of DNA synthesis at induced DSBs provided evidence for some sort of increased POLD3-dependent, and DSB repair-associated, DNA synthesis when SETX was depleted. This increased DNA synthesis was partially dependent upon BLM and inferred to be cytotoxic as it correlates with cell death. Interestingly, examination of cancer data for tandem duplications greater than 100 kb, characteristic of BIR, revealed a slight, but significant, increase in cancers with low SETX but high POLD3 (or BLM) expression. This is consistent with an association between this BIR signature and defective R-loop resolution.

Overall, this study has potential but is somewhat underdeveloped and overstated.

Major Points:

1. The suggestion that R-loop formation is independent of resection appears to be at odds with the recently proposed model of Liu et al (Cell 2021) in which RNA Pol III-dependent RNA synthesis is templated on the exposed 3' ssDNA upon resection at DSBs, forming an RNA:DNA hybrid which transiently protects the 3' ssDNA generated until it is ultimately bound by RAD51. This study should be cited and reconciled with the author's data.

2. The manuscript is lacking somewhat in mechanistic insight and relies overly heavily on one specific cancer cell line (U2OS). Supporting data from an independent system, ideally a non-transformed cell line, would strengthen the interpretations and validate the conclusions drawn.
3. The study requires additional genetic studies to gain further mechanistic insight as to how BLM and POLD3 are functioning in the presence of extensive R-loops. For example, specific mutants defective in either helicase or polymerase activity, rather than just depletion experiments.
4. An independent siRNA, plus cDNA rescue experiments are required to validate key phenotypes.
5. Figures 3 and 4, as currently presented, should be merged into one figure.
6. The title of the manuscript suggests that BLM and POLD3 function together in the same pathway. However, this has not been demonstrated in the results presented. No analysis of the epistatic relationship between BLM and POLD3 is shown.
3. Fig 3A should also include measurement of repair DNA synthesis when SETX is present but BLM is not.
4. Fig 3B should also include measurement of repair DNA synthesis when SETX is present but POLD3 is not.
5. Pg 6, Para 1, Line 7 & Fig1A - Please clearly indicate in the text why the DSBs on Chr.1 and Chr.20 were selected as examples. Are they two similar examples? Do they represent the RNAPII bound and unbound DSBs? This should be clearly stated in the text.
6. Pg7, Para 1 & Fig 1D, S1E – Please define the parameter used in the four categories of RNAPII plotted. If there is a correlation between RNAPII signal and BLM recruitment, is it possible to show a correlation plot rather than a box plot? If not, please clearly indicate how the subset cut-off points were chosen.
7. Pg 9, para 2, lines 3-7 (“...which further supports a role of BLM in promoting a cytotoxic pathway upon...”) - There is no mention of a role of BLM in promoting a cytotoxic pathway previous to this point. Secondly, BLM depletion partially rescues the cytotoxicity observed upon SETX depletion. It is overinterpretation here to suggest that BLM promotes a cytotoxic pathway.
8. Pg 10, para 2, line 2 (“...does not strongly rely...”) the authors state in a previous paragraph that R-loop formation is largely independent of ssDNA. Please clarify precise meaning here.
9. Pg 11, para 2, line (“Our data indicate that an alternative POLD3/BLM-dependent repair pathway occurs at DSBs in SETX-deficient human cells”). This sentence is clearly overstated, in so far as the authors have only shown that both POLD3 and BLM can independently partially rescue the SETX phenotype (see point 2 above), however, they have not shown a POLD3/BLM-dependent pathway.
9. In some parts of the results the authors use the terms “RNA:DNA hybrids” and “R-loops” interchangeably. However, in the discussion they correctly make a clear distinction between RNA:DNA hybrids and R-loops. To avoid confusion, this distinction should be made clearly at the beginning of the manuscript and the terminology strictly used throughout the remainder of the manuscript.

Minor points:

1. Please choose either 4OHT or OHT as an abbreviation for 4-hydroxytamoxifen, not both. Also, as this acronym is for a drug used to induce AsiS1 translocation into the nucleus, it might be easier for the non-specialist reader to consider using ‘+DSBs’ and ‘-DSBs’ in the figures, with a brief explanation in the legend.
2. Please correct punctuation/typos throughout the manuscript.
3. Please cite reference(s) for “previously reported functions of BLM during HR.” (Page 13)
3. Fig S2D, DAPI should be capitalised and the scale bar is missing.
4. Statistical analysis, significance and p-values are missing from a lot of figures, and p-values and

significance should be indicated for all relevant comparisons.

5. Pg 7, para 2, line 3 - Methods sections for cell cycle analysis is missing.

Point by point response to Reviewers comments

Brief summary of the main additions to the revised manuscript

First of all, we would like to thank the three referees for their work on our manuscript. We were glad to see that they all found our study of interest. In our revised manuscript we have included a large number of additional data and analyses. Among the most important additions:

1. We now provide insights on BLM recruitment in SETX-proficient cells (New Fig. 2, Fig. S1B, D; G, Fig. S2): i) We provide evidence that BLM recruitment is favored at guanine quadruplexes (G4) forming loci. ii) We further reinforced the correlation between BLM recruitment and prior RNAPII enrichment at DSBs. iii) We examined the recruitment of BLM across the cell cycle. iv) We analyzed BLM genomic distribution upon sustained DSB induction for 24h which showed considerable extended spreading, in a manner that parallels Rad51. Altogether our data suggest that BLM is preferentially recruited at TC-DSBs, mostly in G2 and that it spreads as resection progresses.
2. To further investigate DNA synthesis at DSBs, we implemented a protocol to analyze DNA synthesis at a genome wide scale (EdU-seq), upon depletion of BLM, or SETX and of combined BLM/SETX depletion (Fig. 4D-E; Fig. S6). In agreement with our EdU-qPCR data (Fig. 4C), we report that DNA synthesis increases at TC-DSBs in SETX deficient cells, and that further depletion of BLM reduces it (Fig. 4D). Moreover, we found that in SETX-depleted cells, the pattern of DNA synthesis correlates with the accumulation of R-loops (Fig. 4E, Fig. S6C). Altogether these new data further reinforce our model that impaired R-loop removal triggers DNA synthesis at TC-DSBs.
3. We found that repair synthesis that takes place at DSB upon RNA:DNA hybrids accumulation is DRB-dependent (Fig. S5C) and Rad51-independent (Fig. S5D).
4. We found that upon SETX depletion, BLM recruitment enhances at DSB (Fig. 4A), in agreement with a function of BLM downstream of R-loop accumulation.
5. We investigated cell survival after TC-DSB induction upon co-depletion of both BLM and POLD3. Co-depletion did not significantly further rescued cells survival in SETX depleted cells, suggesting that BLM and SETX function in the same pathway (Fig. S7E).
6. We further report that POLD1 (the catalytic subunit ensuring DNA synthesis) depletion also improves cell survival in SETX deficient background (Fig. S7D), further establishing POLD3/POLD1-dependent DNA synthesis as a toxic pathway that handles DSBs in absence of R-loop removal.
7. We report that in addition to rescuing cell toxicity observed in SETX-deficient cells, BLM and POLD3 depletions also rescue the increased translocation frequency observed upon SETX depletion (Fig. 5D, Fig. S7F).

Reviewer #1 (Remarks to the Author):

In this manuscript, the authors show that BLM is recruited to DSBs at sites of active transcription. Consistent with the known functions of BLM, depletion of BLM reduces DNA end resection, RAD51 recruitment, and DSB repair accuracy. Because SETX and BLM are similarly recruited to DSBs in active genes, the authors investigated their functional relationship. Interestingly, depletion of BLM rescued the viability of SETX knockdown cells in the presence of DSBs. This rescue by BLM is not attributed to a reduction in resection or R-loops, but

correlated with a reduction in EdU incorporation at DSBs. Depletion of POLD3 and PIF1 also reduced EdU incorporation at DSBs and rescued the viability of SETX knockdown cells after DSB induction, which suggest that BIR is responsible for the toxic DNA synthesis. The authors proposed an interesting model that BLM suppresses the induction of toxic BIR by R-loops. Although the model of this is interesting, the current data are still underdeveloped. Addressing the comments would help strengthen the proposed model.

1. How is BLM preferentially recruited to DSBs in active genes? How does BLM spread 5-10 kb around DSBs? Some mechanistic details would be helpful.

We thank the reviewer for their positive comments on our work. In order to address these two first points, we have now added additional analyses and experiments.

- We now show that BLM recruitment correlates with the presence of active G4 on the genome (detected by the BG4 antibody, see Hänsel-Hertsch 2016, PMID: 7618450) (Fig. S1B, E). This finding is in good agreement with previous work showing that, in BLM deficient cells, Sister Chromatid Exchange (SCE) breakpoints are enriched at genes in general and at G4- carrying genes in particular (van Wietmarschen, et al, 2018, PMID: 29348659)
- We analyzed recruitment of BLM across the cell cycle by ChIP and live imaging of GFP-tagged BLM. We found that BLM is recruited in G1 cells, but its recruitment is enhanced in G2 cells (Fig. S2). Of interest, the difference of BLM recruitment between G1 and G2 was more pronounced when analyzed by ChIP (Fig. S2B) compared to when it was assessed by live imaging (Fig. S2A). This may indicate that BLM is more transiently bound to DSBs in G1 compared to G2, as ChIP assays are particularly sensitive to transient binding.
- We performed additional ChIP-seq to study to distribution of BLM at a later time point. We found that BLM is able to spread on surrounding chromatin on up to 20-40kb at 24h post DSB induction, and in a manner that strongly parallels Rad51 spreading (Fig. 2A-D). This suggests that BLM spreads as resection progresses.

2. Given that BLM is required for efficient DNA end resection and RAD51 recruitment (Fig. S2C, S2D), it is surprising that BLM knockdown did not affect DSB repair efficiency (Fig. S2F). An explanation is needed.

In the revised version we analyzed repair kinetics using an assay previously developed (Fig. S3C). We found a very mild repair delay in BLM deficient cells. We further examined whether an alternative repair pathway could handle TC-DSB in absence of BLM. ChIP against PARP1 revealed an increase of PARP1 recruitment at DSBs in absence of BLM (Fig. S3E) suggesting that Alternative NHEJ is ensuring repair in these conditions. This increased Alt-NHEJ usage is in agreement with the decreased repair fidelity observed in BLM-deficient cells (Fig. S1D), and with a previous report (Grabarz et al, 2013, PMID:24095737)

3. In Fig. 2C, is the helicase activity of BLM required for the reduction of viability in siSETX cells?

In order to address this point, we tried to perform clonogenic assays upon BLM WT and helicase-dead overexpression in D_{IV}A cells depleted for BLM and/or SETX. Unfortunately, overexpression of WT-BLM in D_{IV}A cells triggered a strong decrease in viability precluding further analyses. However, of interest, FANCD1 and RTEL1, two other helicases with an opposite directionality to BLM, but displaying, as BLM, helicase activity toward G4 structures, also rescued cell survival upon SETX depletion (Fig. for Referee. 1). This may suggest a more general involvement of G4 resolvases in promoting POLD3 –dependent DNA synthesis and rescuing the lethality observed in SETX-deficient following TC-DSB induction. We could include these data in the manuscript if required.

Fig. for referee 1

Clonogenic assays were performed in AID-D_{IV}A cells, following transfection with SETX and/or FANCD1 (top) or RTEL1 (bottom) siRNA as indicated. Values are mean +/-SEM (n=4 (top panel, or n=3 bottom panel) and P values are indicated (paired t-test).

4. In Fig. 2D, although siBLM cells are defective in resection, siSETX+siBLM cells have more R-loops than siSETX cells. Why would siSETX increase R-loops more in siBLM cells? Should one expect that siBLM reduces R-loop levels even in the absence of SETX?

We thank the reviewer for raising this point. We now shown in Fig. S4D a quantification of R-loops (DRIP-seq read count) at the 80 best induced DSB upon Ctrl, SETX, BLM and SETX+BLM siRNA transfection. The slight decrease observed on average profile between cells transfected with Ctrl and BLM siRNA (Fig. 3D, grey and blue respectively) was not significant (Fig. S4D). This shows that decreased resection (in BLM depleted cells) does not translate into decreased R-loop formation, and hence suggests that single strand generation is not required for R-loop accumulation at TC-DSBs, a result further reinforced by the lack of correlation between

R-loops and Rad51 distribution at later time points (Fig. 3E). This supports a model where following DSBs, R-loops form on dsDNA upstream of resection.

Along the same line, BLM depletion in SETX-deficient cells, also did not decrease R-loop accumulation (Fig. 3D, red and purple lines respectively). Rather, as this referee noticed, BLM deficiency in SETX-deficient cells even further increased R-loop accumulation at DSBs. This difference was significant (Fig. S4D red and purple boxplots respectively, $P=6e-8$). The reason for such an exacerbated R-loop accumulation upon co depletion of BLM and SETX is yet unclear, but we could envisage two possibilities: first, given that BLM displays a G4 helicase activity, BLM depletion may trigger G4 accumulation at the vicinity of DSBs, giving more opportunity for R-loops to form and accumulate, especially in a context where they can't be removed (SETX depletion). Alternatively, the slight delay in repair kinetics upon BLM depletion may give more time for DSB-induced R-loops to accumulate, which would be particularly visible in a background deficient for R-loop-removal.

In any case, our data indicate that BLM depletion does not rescue the R-loop accumulation observed upon SETX depletion, while it does decrease resection and rescues cell survival. This indicates that i) R-loop accumulation is not altered by impaired resection and that ii) the ability of BLM to rescue cell survival in SETX-deficient cells is not due to increased clearance or decreased accumulation of R-loop at DSBs. Altogether this places BLM contribution in promoting cell death in SETX-deficient background downstream of R-loop accumulation.

5. In Fig. 3A, can the authors test if the EdU incorporation at DSBs is RAD51 and/or R-loop dependent?

We have now performed EdU-qPCR upon siRNA against Rad51 (Fig. S5D). We found that Rad51 is not involved in DNA synthesis that takes place at DSB in absence of SETX. Moreover, DRB treatment rescued the strong increase of DNA synthesis observed at TC-DSBs in SETX deficient cells (Fig. S5C) suggesting that DNA synthesis depends on R-loop accumulation upon SETX depletion.

6. In Fig. 3, are BLM and POLD3 'epistatic' for the EdU incorporation in siSETX cells?

To address the epistasis between BLM and POLD3 in SETX-deficient cells, (a point also raised by Reviewer #3) we have now performed clonogenic assays upon depletion of BLM, SETX and POLD3. We found that POLD3/BLM co depletion did not further significantly rescue cell survival in SETX-deficient cell compared to POLD3 alone (Fig. S7E), suggesting that BLM and POLD3 function in the same pathway.

7. Although the EdU incorporation at DSBs in siSETX cells correlates with the reduction in cell survival, there is no direct evidence that this DNA synthesis is responsible for the reduction of cell viability. Although disruption of BIR reduces EdU incorporation at DSBs and rescues the viability of siSETX cells, these two effects may not be causally linked. This is an important issue to address. Any evidence that BIR generates toxic DNA structures or secondary DNA damage?

We agree with this referee that we did not demonstrate at a molecular level the relationship between BIR usage and cell clonogenic potential. We tried to be careful in the manuscript to not over-interpret our data. Of interest in yeast, Pol32 (POLD3 ortholog) and PIF1 usage in *rnaseh1/2* deficient strain is also associated with a strong decrease in viability (Amon et al, 2016, PMID:.27938663). This loss of viability was not due to checkpoint activation (Amon et al, 2016, PMID:.27938663; Costantino et al, PMID: 30078723). Importantly, multiples studies identified BIR as responsible for genomic instability (clustered mutations, gross chromosome rearrangements) in both yeast and mammalian cells (e.g. Sakovsky et al, 2014 PMID: 24882007; Pardo et al, 2012; PMID: 23071463; Hastings et al, 2009, PMID: 19180184). We can postulate that the increase usage of BIR when multiple breaks are induced simultaneously, may translate into a large amount of chromosome instability, impaired chromosomal segregation and cell death.

To further address this, we now analyzed the effect of POLD3 depletion on translocation frequency, that we previously found to be strongly increased upon SETX depletion (Cohen et al, 2018, PMID: 29416069). Importantly, both POLD3 and BLM depletion rescued the high level of translocation observed in SETX-deficient cells (Fig. 5D, Fig. S7F), suggesting that a POLD3-dependent pathway is at least in part responsible for the occurrence of translocations upon impaired R-loop removal. Of note FANCI depletion also rescued this increase of translocation frequency (see below Fig. for referee. 2), in agreement with its ability to rescue impaired cell survival upon R-loop accumulation (Fig. for referee. 1). Moreover, we also report that the depletion of the catalytic subunit POLD1 of POL δ improves cell survival in SETX-deficient cells, further linking DNA synthesis to cell death (Fig. S7D). Altogether these data show that POLD3, which is responsible for DNA synthesis at TC-DSBs upon R-loop accumulation promotes both cell death and an increased translocation rate. We also discuss this point in the discussion section (p17).

Fig. for referee. 2

Two translocations were analyzed in DivA cells, upon transfection with the indicated siRNA. Mean and SEM of n=4 is shown P values are indicated (paired t-test).

8. The data in Fig. 4 seem underdeveloped. What is the basis to think that BIR is the primary cause of TD <100kb? Although the paper by Costantino et al. suggested that BIR can induce TD, it may or may not be the primary cause of TD in cancer cells. Also, the functional meaning of high or low levels of SETX, POLD3, and BLM is unclear. Does high POLD3 or BLM mean more BIR activity? Are POLD3 and BLM overexpressed in tumors or simply not down regulated? When SETX is low, is it low enough to increase R-loops? It seems to me that the interpretations of the data in Fig. 4 are based on a number of untested presumptions. It will take more work to substantiate these interpretations.

We apologize if we implied that BIR is the primary cause of TD in cancer cells. Our aim was not to show a direct causal link between the expression levels of BLM, POLD3 and SETX and the genomic instability in cancer cells. Rather the purpose here was to use cancer genomic databases as a mean to investigate whether a genetic interaction between SETX and POLD3 or BLM could also be observed in an *in vivo* context. Indeed, we could observe such a genetic interaction since SETX low expression was associated with more TD, unless if POLD3 or BLM expression was low. A similar approach was used to show a genetic interaction between DYNLL1 and BRCA1 (He et al, 2018, PMID: 30464262). In order to make clear and to avoid over-interpretation, we have now rephrased.

Of note, expression levels for BLM, POLD3 but also POLD1 indeed show overexpression in pancreatic cancer samples (TCGA) compared to normal tissues (Fig. for referee 3). We could include these data in the manuscript upon request.

Fig. for referee. 3

Expression of BLM, POLD3 and POLD1 was plotted in pancreatic adenocarcinoma (using TCGA datasets, n=179) and compared to TCGA normal + GTEx datasets (n=171) using GEPIA2. * P<0.01.

Reviewer #2 (Remarks to the Author):

This is a review of NCOMMS-21-14809. In this study, the authors examine the action of the BLM helicase at chromosomal double strand breaks (DSBs) induced by the nuclease AsiSI. Data are presented that BLM is recruited to a greater degree to DSBs in transcribed loci, the R-loop dissolution factor SETX is important to suppress nascent DNA synthesis at DSBs (EdU labeling / IP method), and that such nascent DNA is promoted by BLM and the Polymerase delta subunit POLD3. The reduction in EdU labeling in the BLM+SETX co-depleted cells vs. SETX depleted alone correlates with partial rescue of viability in the co-depleted vs. SETX depleted alone. Hence, the authors conclude that BLM-mediated DNA synthesis in SETX-depleted cells is toxic. Controls showing these effects of BLM on lethality in SETX-deficient cells are independent of end resection and RAD51-loading are shown. Altogether, the study indicates that DSBs with R-loops are prone to DNA synthesis that is consistent with a break-induced replication mechanism, as found in ALT telomere maintenance. A lot of the data are convincing, but throughout there are many missing controls, and some data that is not as convincing. There are also some issues with interpretation.

Major points: I found the most novel/compelling part of the study to be the EdU labeling at DSBs, but this seems somewhat underdeveloped, as follows.

1. The effect of BLM and POLD3 depletion on EdU labeling is not shown (Fig 3A/B). Presumably, in the presence of SETX, EdU labeling is weak, and as such any role in BLM and POLD3 in promoting these events may not be detectable. However, this should be directly shown, and furthermore the possibility that BLM and POLD3 suppress EdU labeling (in SETX+ cells) should be formally ruled out with this experiment.

First of all we thank the referee for their work and appreciation on our manuscript. To answer these first points, we now show the EdU incorporation at DSBs in SETX proficient cells upon depletion of BLM (Fig. S5E) and POLD3 (Fig. S7B) alone. BLM depletion did not affect the detectable EdU incorporation in SETX-proficient cells. POLD3 depletion alone on another hand increased EdU incorporation at DSB in SETX-deficient cells. The meaning for increased repair synthesis in absence of POLD3 is unclear but may reflect an inhibitory role of POLD3 on homologous recombination repair in cells proficient for R-loop removal.

As for a potential suppression of general EdU labeling in BLM, SETX, POLD3 or combined BLM/SETX and POLD3/SETX depleted cells, we provide below (Fig. for referee 4) a figure showing that all depletions do not affect EdU incorporation. When compared to mimosine-treated cells (when replicative DNA synthesis is inhibited), EdU incorporation at a replication origin in cycling cells was not dramatically affected by the depletion of SETX, BLM or both combined (panel A, a single experiment is shown, technical qPCR replicates, raw data are presented as % of input). Variations are Ori can be due to pull down efficiency from one point to another, which is the reason why qPCR analysis of DNA synthesis at DSBs was normalized against qPCR data obtained detected at origins (as often performed in ChIP experiment when a control genomic locus is used as a normalization factor, or in RT-qPCR experiments when qPCR are normalized against a reference cDNA such as RPLP0 or ACTB).

Across 5 biological independent experiments, BLM, SETX, POLD3 or their combination did not trigger significant changes of EdU level in replicative cells at the replication origin (panel B, average raw data collected from N=5 are presented as % of input). We could include these controls in the manuscript upon request.

Fig. for referee. 4

A. qPCR data obtained at an origin of replication, following Edu-pull down performed in DivA cells transfected with control (Ctrl), SETX, BLM and SETX+BLM siRNA as indicated and treated with OHT or not to induce DSBs. EdU-pull down was performed in asynchronous growing cells ongoing replicative DNA synthesis, or in cells where replicative DNA synthesis is inhibited. A single experiment (qPCR triplicates) is shown.

B. qPCR data obtained at an Ori across 5 independent experiments in cells transfected with the indicated siRNA showing that EdU labelling at an Origin is not affected by siRNA treatment.

2. There is clearly much more to learn about the EdU incorporation at DSBs caused by SETX depletion (what is the template? why are R-loops causing such an induction of EdU labeling? what other transcription / repair defects trigger such EdU incorporation?), and certainly more that can fit in the first report, but two comparisons relevant to this study seem in order a) is the synthesis in SETX-depleted cells affected by RAD51-depletion? b) An assumption is that SETX is acting in cis at these DSBs to clear R-loops and thereby suppress EdU labeling. This could be assessed through at least one of these control experiments: does DRB treatment also cause loss of the EdU-labeling? does RNaseH expression cause a loss in the EdU-labeling?

To answer these points we have now performed EdU-qPCR upon siRNA against Rad51 (Fig. S5D). We found that Rad51 is not involved in DNA synthesis that takes place at DSB in absence of SETX. Moreover, DRB treatment rescued the strong increase of DNA synthesis observed at TC-DSBs in SETX-deficient cells (Fig. S5C) suggesting that DNA synthesis depends on R-loop accumulation upon SETX depletion.

Moreover, in order to go further and gain more insights into the DNA synthesis that takes place at TC-DSBs we implemented a genome wide assay (EdU-seq) to analyze DNA synthesis, upon depletion of BLM or SETX and of combined BLM/SETX (Fig. 4D-E; Fig. S6). In agreement with our EdU-qPCR data (Fig. 4C), we report that DNA synthesis increases at TC-DSBs in SETX-deficient cells, and that further depletion of BLM reduces it (Fig. 4D). Moreover, we found that in SETX-depleted cells, the pattern of DNA synthesis correlates with the accumulation of R-loops (Fig. 4E, Fig. S6C). Altogether these new data further reinforce our model that impaired R-loop removal triggers DNA synthesis at TC-DSBs.

Other points:

3. The degree to which BLM recruitment to DSBs is affected by transcription level at the locus should be described more rigorously. Namely, while there appears to be a pattern favoring BLM recruitment to transcribed loci, the affect seems relatively mild, particularly in comparison to how RAD51 is recruited. A more rigorous description, which appears to include evidence for BLM recruitment to DSBs in non-transcribed loci too, will improve the lasting impact of the study.

We provide individual examples (Fig. S1H) that show difference of BLM recruitment between transcribed loci (DSB1-DSB4) and a DSB lying in a non-transcribed locus as assessed by RNA-seq. Moreover, the averaged profiles of RNAPII and RNAPIIS2P at all 80 best DSB induced in DIvA cells show difference between BLM-high and BLM-low DSBs, typical of active transcription (Fig. S1C, *i.e.*: an accumulation RNAPIIS2 at 3' ends and an accumulation of total RNAPII at promoters for the BLM-high subset). The reverse was also true, when DSB were sorted based on RNAPII, or RNAPIIS2, those highly enriched in RNAPII recruited more BLM (Fig. 1D, Fig. S1F). In order to further establish the correlation between RNAPII and BLM at DSBs we now also show scatterplots showing BLM and RNAPII read count from CHIP-seq performed in DIvA cells (Fig. S1G). In agreement with this correlation obtained using different analyses, prior DRB treatment did decrease BLM recruitment at DSB. Altogether, while we do not imply that BLM binds only at TC-DSBs, we believe that we provide a strong body of evidence that BLM binding is biased toward TC-DSBs (*i.e.* in the cell population a given DSB has more chance to recruit BLM if it is in a RNAPII-bound locus than if not).

4. Along the lines of point 1, Figure 1E doesn't have statistical analysis.

We now included the statistical analysis

5. A key part of their model is that SETX loss should probably cause elevated BLM recruitment, but this is not tested. Even the negative result would be informative and important for

We thank the referee for their suggestion. We have now tested BLM recruitment by ChIP in cells depleted for SETX. We could detect increased BLM binding at TC-DSBs in SETX-deficient cells (Fig. 4A)

6. As Nature Comms is a long-formal journal, please more many of the Supplemental Figure panels to main figures, particularly Fig S4, if not more of them.

The Figures have now been modified to include more data.

Reviewer #3 (Remarks to the Author):

Cohen et al map the BLM helicase to the vicinity (5-10 kb) of DSBs using their well characterised DlvA system. DSBs at actively transcribing genes were particularly enriched in BLM and this enrichment was reduced upon inhibition of transcription. Mechanistically, they demonstrate that BLM is required for resection of DSBs. Previously, these authors have shown similar localisation of SETX, which can resolve R-loops, to DSBs at actively transcribed loci. Surprisingly, BLM depletion partially rescued cell death observed upon depletion of SETX due to defective removal of DNA:RNA hybrids. The authors speculate that as BLM promotes long-range resection, its depletion may rescue DSB-induced cell death in SETX depleted cells by reducing resection-dependent DNA:RNA hybrid formation. Surprisingly, measurement of R-loop formation by DRIP-seq did not reveal significant difference in their formation upon BLM depletion, suggesting that R-loop formation is independent of resection (see major point 1, below).

Unlike BLM, depletion of CtIP, which is required for initiating resection, did not rescue the lethality observed upon DSB induction in SETX-depleted cells. The authors have interpreted this as evidence for BLM promoting a cytotoxic pathway upon the excessive R-loop formation upon SETX depletion. Measurement of DNA synthesis at induced DSBs provided evidence for some sort of increased POLD3-dependent, and DSB repair-associated, DNA synthesis when SETX was depleted. This increased DNA synthesis was partially dependent upon BLM and inferred to be cytotoxic as it correlates with cell death. Interestingly, examination of cancer data for tandem duplications greater than 100 kb, characteristic of BIR, revealed a slight, but significant, increase in cancers with low SETX but high POLD3 (or BLM) expression. This is consistent with an association between this BIR signature and defective R-loop resolution.

Overall, this study has potential but is somewhat underdeveloped and overstated.

Major Points:

1. The suggestion that R-loop formation is independent of resection appears to be at odds with the recently proposed model of Liu et al (Cell 2021) in which RNA Pol III-dependent RNA synthesis is templated on the exposed 3' ssDNA upon resection at DSBs, forming an RNA:DNA hybrid

which transiently protects the 3' ssDNA generated until it is ultimately bound by RAD51. This study should be cited and reconciled with the author's data.

We thank the referee for their work and appreciation on our manuscript. We also thank them for raising this important point. Indeed, we provide several lines of evidence that R-loop formation, at least detected by our approach using the S9.6 antibody, is not a direct consequence of single strand DNA availability.

- While BLM depletion in a SETX-deficient background reduces resection compared to Ctrl cells or SETX-depleted cells (Fig. S4B), it does not reduce R-loop formation (Fig. 3D; Fig. S4D)
- While Rad51 spreads on large domains upon longer DSB induction as long range resection progresses, this is not the case for R-loop accumulation (Fig. 3E)
- CtIP depletion, while decreasing resection, does not rescue the SETX phenotype (Fig. S4E-F)

Our data are not necessarily at odds with the Liu et al paper (PMID: 33626331, cited in our manuscript). This paper did not suggest that RNA Pol III-dependent RNA synthesis is templated on the exposed 3' ssDNA, but rather that once synthesized, the RNA further hybridizes with the available single stranded DNA. They suggest a role of hybrids in promoting resection, and place RNA formation upstream DNA end resection, which agree with our data. This is now discussed p16.

Yet, indeed, our data disagree with the model proposed from various studies in which these long non-coding RNAs initiated from the DNA ends (either from RNAPII or RNAPIII-mediated transcription, a yet controversial point) further hybridize on the entire length of available single stranded DNA. However, there are no data to date that formally demonstrated this model, since it requires to analyze R-loop formation using a high resolution, genome wide and strand specific assay (such as strand specific qDRIP-seq). Our DRIP-seq invalidate this model and show that, unlike Rad51, R-loops do not accumulate farther as resection progresses.

Furthermore, while this falls beyond the scope of this paper, we have thoroughly investigated in our laboratory, RNAPII and RNAPIII recruitment at DSBs, using ChIP-seq to avoid the confounding effect of RNA polymerase immobilization (for instance a DSB-induced change of RNAPII elongation or release from chromatin) that could look on laser stripes as if RNA Pol are recruited. We could not observe either RNAPII or RNAPIII *de novo* recruitment (Fig. for referee 5, A and B respectively). Consequently, we rather favor the hypothesis that RNA:DNA hybrids arise at TC-DSBs due to transcriptional repression (known in other context to induce R-loop formation). In conclusion on this important point, we believe that additional work is needed to understand exactly the nature of RNA:DNA hybrids that form at DSBs and to substantiate the model previously proposed of *de novo* formed-lncRNA hybridized with ssDNA.

Fig. for referee. 5

A. BLESS, RNAPII ChIP-seq, and RNA-seq genomic tracks in presence or absence of DSB as indicated, at a genomic locus showing active transcription (left) or not (right). DSB are indicated by arrows. No recruitment is seen, despite i) clear DSB induction (see BLESS), ii) efficient RNAPII ChIP-seq (see green signal on gene). Rather we can detect decreased RNAPII post-DSB induction, in agreement with previously reported transcriptional inhibition (also shown here by RNA-seq). B. POL3RA and POL3RE ChIP-seq (two subunits of RNAPIII) at a control locus (left panel annotated tRNA) and at a DSB (right panel). No RNAPIII recruitment is observed at DSB despite efficient ChIP-seq.

2. The manuscript is lacking somewhat in mechanistic insight and relies overly heavily on one specific cancer cell line (U2OS). Supporting data from an independent system, ideally a non-transformed cell line, would strengthen the interpretations and validate the conclusions drawn.

Unfortunately, at this stage, we have not been successful yet to develop other non-cancer and cancer cell lines, expressing the AID-DivA system (the auxin degron being essential for clonogenic assays, but a difficult construct to establish and maintain in cells), this work is still under progress. Of importance, U2OS are ALT-positive cells. Given that ALT positive cells utilize a BIR like pathway to maintain their telomeres, it is possible that SETX deficiency is particularly toxic following TC-DSBs in ALT –positive cells (which may over use a POLD3-dependent pathway). Investigating BLM/POLD3 and SETX interplay in ALT-positive and ALT-negative will deserve further investigations. We now mention this point in the discussion. Nevertheless, our analysis using cancer genome database, revealed that such a genetic interaction between POLD3 and SETX or BLM and SETX, could also be seen in an *in vivo* context, suggesting that our finding can be generalized to some extent.

3. The study requires additional genetic studies to gain further mechanistic insight as to how BLM and POLD3 are functioning in the presence of extensive R-loops. For example, specific mutants defective in either helicase or polymerase activity, rather than just depletion experiments.

For POLD3 polymerase dead, POLD3 does not carry a DNA synthesis catalytic activity, but we could show that POLD1 (the catalytic subunit of POL δ) depletion also similarly rescues cell survival. This data is now presented Fig. S7D.

Unfortunately, we have not been able to perform WT and helicase-dead BLM rescue experiments since overexpression of WT-BLM in DivA cells triggers extensive cell death. However, we observed that other G4 DNA helicases (FANCI and RTEL1) behave similarly to BLM, in that they also rescue cell death observed upon SETX depletion as well as translocations (see above Fig. for referee 1-2).

4. An independent siRNA, plus cDNA rescue experiments are required to validate key phenotypes.

We performed experiment using a second siRNA against POLD3 (Fig. S7C) and BLM (Fig. S4A). As mentioned above BLM WT overexpression triggered extensive cell death in DivA cells so we could not further perform cDNA rescue experiments.

5. Figures 3 and 4, as currently presented, should be merged into one figure.

We have now reordered the figures. A Part of old Fig. 3, and old Fig. 4 are now in Fig. 5

6. The title of the manuscript suggests that BLM and POLD3 function together in the same pathway. However, this has not been demonstrated in the results presented. No analysis of the epistatic relationship between BLM and POLD3 is shown.

To address the epistasis between BLM and POLD3 in SETX-deficient cells, (a point also raised by Reviewer #1) we have now performed clonogenic assays upon depletion of BLM, SETX and POLD3. We found that POLD3/BLM co depletion did not further significantly rescued cell survival in SETX-deficient cell compared to POLD3 alone (Fig. S7E), suggesting that BLM and POLD3 function in the same pathway.

3. Fig 3A should also include measurement of repair DNA synthesis when SETX is present but BLM is not.

This has been done and is now presented Fig. S5E

4. Fig 3B should also include measurement of repair DNA synthesis when SETX is present but POLD3 is not.

This has been done and is now presented Fig. S7B

5. Pg 6, Para 1, Line 7 & Fig1A - Please clearly indicate in the text why the DSBs on Chr.1 and Chr.20 were selected as examples. Are they two similar examples? Do they represent the RNAPII bound and unbound DSBs? This should be clearly stated in the text.

We apologize if this was not clear, they just represent two DSBs that we know are well cleaved in our system (Clouaire et al, Mol Cell, 2018), both are in a transcribed region. We now mentioned this in the text. Of note, the DSB on the top panel of Fig. 1A is the DSB presented on Fig. for referee 5A left. And the DSB presented on the bottom panel is DSB-3 presented in Fig. S1H.

6. Pg7, Para 1 & Fig 1D, S1E – Please define the parameter used in the four categories of RNAPII plotted. If there is a correlation between RNAPII signal and BLM recruitment, is it possible to show a correlation plot rather than a box plot? If not, please clearly indicate how the subset cut-off points were chosen.

There were no cut off, the 80 DSBs were just separated in 4 equal categories (20 each) based on their RNAPII signal. This is specified in the methods section p.23. We now also show scatterplots (Fig. S1G)

7. Pg 9, para 2, lines 3-7 (“...which further supports a role of BLM in promoting a cytotoxic pathway upon...”) - There is no mention of a role of BLM in promoting a cytotoxic pathway previous to this point. Secondly, BLM depletion partially rescues the cytotoxicity observed upon SETX depletion. It is overinterpretation here to suggest that BLM promotes a cytotoxic pathway. We have now rephrased and do not mention BLM as promoting a cytotoxic pathway

8. Pg 10, para 2, line 2 (“...does not strongly rely...”) the authors state in a previous paragraph that R-loop formation is largely independent of ssDNA. Please clarify precise meaning here. We have now rephrased to be more clear about what we meant.

9. Pg 11, para 2, line (“Our data indicate that an alternative POLD3/BLM-dependent repair pathway occurs at DSBs in SETX-deficient human cells”). This sentence is clearly overstated, in so far as the authors have only shown that both POLD3 and BLM can independently partially rescue the SETX phenotype (see point 2 above), however, they have not shown a POLD3/BLM-dependent pathway.

This was addressed (point 6)

9. In some parts of the results the authors use the terms “RNA:DNA hybrids” and “R-loops” interchangeably. However, in the discussion they correctly make a clear distinction between RNA:DNA hybrids and R-loops. To avoid confusion, this distinction should be made clearly at the beginning of the manuscript and the terminology strictly used throughout the remainder of the manuscript.

We now mentioned in the introduction the terminology. We have decided to only use the term RNA:DNA hybrids throughout the manuscript, but we do not know yet whether R-loops or RNA:DNA hybrids form at DSBs (see point 1, it would require strand specific DRIP-seq). Our data suggest that R-loop form, rather than RNA:DNA hybrids. This is discussed p.17.

Minor points:

1. Please choose either 4OHT or OHT as an abbreviation for 4-hydroxytamoxifen, not both. Also, as this acronym is for a drug used to induce AsiS1 translocation into the nucleus, it might be easier for the non-specialist reader to consider using ‘+DSBs’ and ‘-DSBs’ in the figures, with a brief explanation in the legend.

Thanks for the suggestion, this makes the figures clearer, we have replaced OHT by +DSB.

2. Please correct punctuation/typos throughout the manuscript.

We proofread the manuscript many times. We hope all have been corrected.

3. Please cite reference(s) for “previously reported functions of BLM during HR.” (Page 13)

References are now cited.

3. Fig S2D, DAPI should be capitalised and the scale bar is missing.

This has been done.

4. Statistical analysis, significance and p-values are missing from a lot of figures, and p-values and significance should be indicated for all relevant comparisons.

This has been done.

5. Pg 7, para 2, line 3 - Methods sections for cell cycle analysis is missing.

This has been done.

REVIEWERS' COMMENTS

Reviewer #1 (Remarks to the Author):

The authors have done a great job in addressing my comments. The manuscript is ready for publication.

Reviewer #2 (Remarks to the Author):

The authors have addressed my concerns with excellence.

Reviewer #3 (Remarks to the Author):

The great efforts of the authors to address all the reviewers comments are appreciated. While they could not experimentally address all queries, the authors have included a substantial amount of new data that has added very significantly to the overall quality of the study.

Point by point response to Reviewers comments

Reviewer #1 (Remarks to the Author):

The authors have done a great job in addressing my comments. The manuscript is ready for publication.

Reviewer #2 (Remarks to the Author):

The authors have addressed my concerns with excellence.

Reviewer #3 (Remarks to the Author):

The great efforts of the authors to address all the reviewers comments are appreciated. While they could not experimentally address all queries, the authors have included a substantial amount of new data that has added very significantly to the overall quality of the study.

We thank all the referees for their work on the manuscript and for their very positive comments.